# Assessment of the data assimilation framework for the Rapid Refresh Forecast System v0.1 and impacts on forecasts of a convective storm case study

Ivette H. Banos[1,2], Will D. Mayfield[2], Guoqing Ge[3], Luiz F. Sapucci[4], Jacob R. Carley[5], and Louisa Nance[2]

[1]Graduate Program in Meteorology, National Institute for Space Research, São José dos Campos, São Paulo, Brazil
[2]National Center for Atmospheric Research, Boulder, CO, USA
[3]NOAA Global Systems Laboratory, and Cooperative Institute for Research in Environmental Sciences, CU Boulder, Boulder, CO, USA
[4]Center for Weather Forecasts and Climate Studies, National Institute for Space Research, São Paulo, Brazil
[5]NOAA/NCEP Environmental Modeling Center, College Park, MD, USA

**Correspondence:** Ivette Hernández Baños (ivette@ucar.edu)

**Abstract.** The Rapid Refresh Forecast System (RRFS) is currently under development and aims to replace the National Centers for Environmental Prediction (NCEP) operational suite of regional and convective scale modeling systems in the next upgrade. In order to achieve skillful forecasts comparable to the current operational suite, each component of the RRFS needs to be configured through exhaustive testing and evaluation. The current data assimilation component uses the hybrid three dimensional ensemble–variational data assimilation (3DEnVar) algorithm in the Gridpoint Statistical Interpolation (GSI) system. In this study, various data assimilation algorithms and configurations in GSI are assessed for their impacts on RRFS analyses and forecasts of a squall line over Oklahoma on 4 May 2020. A domain of 3 km horizontal grid-spacing is configured and hourly update cycles are performed using initial and lateral boundary conditions from the 3 km grid High Resolution Rapid Refresh (HRRR). Results show that a baseline RRFS run is able to represent the observed convection, but with stronger cells and large location errors. With data assimilation, these errors are reduced, especially in the 4 and 6 h forecasts using 75 % of the ensemble background error covariance (BEC) and 25 % of the static BEC with the supersaturation removal function activated in GSI. Decreasing the vertical ensemble localization radius from 3 layers to 1 layer in the first 10 layers of the hybrid analysis results in overall less skillful forecasts. Convection is greatly improved when using planetary boundary layer pseudo-observations, especially at 4 h forecast, and the bias of the 2 h forecast of temperature is reduced below 800 hPa. Lighter hourly accumulated precipitation is predicted better when using 100 % ensemble BEC in the first 4 h forecast, but heavier hourly accumulated precipitation is better predicted with 75 % ensemble BEC. Our results provide insight into current capabilities of the RRFS data assimilation system and identify configurations that should be considered as candidates for the first version of RRFS.

# 1 Introduction

The increase in computational resources over the last several decades has allowed a considerable increase in horizontal reso-
lution in numerical weather prediction (NWP) (e.g., Bauer et al., 2015; Yano et al., 2018). Currently, many NWP centers have developed and use high resolution models operationally for short range weather forecast guidance (e.g., Bannister et al., 2020). These models have provided more realistic forecasts of hazardous weather events where deep convection is explicitly resolved (e.g., Lean et al., 2008). Typically, in models with grid spacing less than 4 km, the deep cumulus parameterization is turned off and convection is treated explicitly, though not necessary completely resolved. Such configurations are therefore often called convection-allowing models (e.g., Schwartz and Sobash, 2019).

The current suite of operational convection-allowing models at the National Centers for Environmental Prediction (NCEP) consists of multiple dynamical cores and physics schemes, none of which have many shared components with their global counterpart, the Global Forecast System (GFS). At present, convection-allowing forecasts are produced by the North Ameri-
can Mesoscale Forecast System (NAM) 3 km nests, High Resolution Rapid Refresh (HRRR), and the High Resolution Window (HIRESW) systems. These systems are then combined into a convection-allowing ensemble known as the High Resolution En-
semble Forecast system (HREF; Roberts et al., 2020). The global modeling suite is based on the Finite-Volume Cubed-Sphere (FV3) dynamical core with a physics suite developed and tuned for global applications, while the regional operational models are based on unique physics suites and dynamical cores, such as the Advanced Research Weather Research and Forecasting model (WRF-ARW; Skamarock et al. (2008)) and Non-hydrostatic Multiscale Model on the B-grid (Janjić et al., 2001).

Considerable human and computing resources and efforts are required to maintain and improve such a variety of models in order to continuously provide successful numerical guidance for different sectors of society (Link et al., 2017). Therefore, the National Oceanic and Atmospheric Administration (NOAA) is currently transitioning toward the Unified Forecast Sys-
tem (UFS; https://ufscommunity.org/) (EMC, 2018). A unified forecasting system brings together advanced developments in weather and climate models, maximizing collective efforts and resources, while also connecting expertise across the scientific community (e.g., Hazeleger et al., 2010; National Research Council, 2012; Brown et al., 2012). Within the UFS framework, the GFS was coupled with the WAVEWATCH III wave model in the operational upgrade of March 2021 (NWS, 2021). The UFS application for convection-allowing forecasts is the Rapid Refresh Forecast System (RRFS) (Alexander and Carley, 2020). RRFS is under development and aims to facilitate the unification of the regional convection-allowing suite of models by sub-
suming the present suite of multi-dynamic core modeling applications in the next operational upgrade (UFS-R2O, 2020).

The FV3 dynamical core developed at the Geophysical Fluid Dynamics Laboratory (GFDL) (Lin, 2004; Putman and Lin, 2007; Harris and Lin, 2013) was selected for UFS applications after a thorough evaluation process (Ji and Toepfer, 2016). In the past several years, multiple studies have been conducted using the FV3 dynamical core for convective scale NWP where it has demonstrated skill (e.g., Potvin et al., 2019; Zhang et al., 2019; Snook et al., 2019; Zhou et al., 2019; Harris et al., 2019, 2020b; Gallo et al., 2021; Black et al., 2021). For example, the grid stretching capability of an FV3-based global model (Harris and Lin, 2013) was evaluated in Zhou et al. (2019). Small scale structures of the convective activity in a squall line case were correctly resolved, although an overprediction of the precipitation and radar reflectivity values was observed. In the

framework of the 2018 NOAA Hazardous Weather Testbed Spring Forecasting Experiment, Gallo et al. (2021) discussed the strengths as well as elements that need improvement in FV3-based convection-allowing models when compared to HRRR, highlighting the overproduction of high reflectivity values (45 dBZ) in storms. A limited area model (LAM) capability based on the FV3 dynamical core (FV3 LAM) has also been developed, which reduces required computational resources associated with having to run a global model to accommodate a nest. Month long tests at convection-allowing resolution with FV3 LAM show comparable performance relative to a two-way nested domain at forecast lead less than 24 hours (Black et al., 2021). Additionally, developments on the UFS hurricane application using the FV3 LAM, the Hurricane Analysis and Forecast System (HAFS), have shown improvements of track and intensity forecasts compared to GFS (Dong et al., 2020).

The RRFS is presently being built upon the UFS Short Range Weather Application (SRW) (Alexander and Carley, 2020). The first version (v1.0.0) of the SRW (UFS Development Team, 2021) was released on March 2021 and it includes the FV3 LAM with pre-processing utilities, the Common Community Physics Package (CCPP), the Unified Post Processing System (UPP), and a workflow to run the system on a variety of high performance computing platforms as well as one's own personal laptop (Wolff and Beck, 2020). Harrold et al. (2021) investigated how the SRW represents convection and associated precipitation for varied model grid spacing in two physics suites: a suite based on GFS version 16 physical parameterizations and a prototype of the RRFS physics suite (henceforth called RRFS_PHYv1a). It was found that for both physics suites a 3 km resolution yields a more realistic representation of convection but with a cool 2 m temperature bias and an underforecast of low reflectivity values. Kalina et al. (2021) also examined these two SRW physics suites and demonstrated that they failed to depict trailing stratiform precipitation in simulations of a squall line and Hurricane Barry (July 2019). Preliminary results indicate that this issue could be related to fewer ice crystals in the model runs than in the radar-derived data. Moreover, the same experimental configuration was used by Newman et al. (2021) to investigate the land-atmosphere interactions using a heatwave case and a winter cold air outbreak case. A cooler planetary boundary layer (PBL) with increased cloudiness and less surface downward shortwave were found in the heatwave case simulations, while an increase is seen in the 10 m wind speed in the cold air outbreak case.

NWP is an initial value problem and convection-allowing forecasts are no different. Forecasts at such scales strongly depend on the quality of the initial conditions and the ability of the analysis algorithm to provide accurate state estimates of fine-scale spatiotemporal structures that are of inherent interest in convection-allowing NWP, such as ongoing convection, complex circulations associated with subtle boundaries (e.g. dry lines), etc. To achieve such analyses with reasonable fidelity, dense and accurate observations are needed in the data assimilation window. However, implementing observation operators for the most dense observation types is often complex, such as radar reflectivity, as they are often indirectly related to state variables. In addition, nonlinear model processes along with non-Gaussian error characteristics are commonplace at the convective scale, both of which encumber the accurate specification of error covariance matrices and, to varying degrees, violate some of the underlying parametric assumptions that are at the foundation of most state-of-the-art analysis algorithms (e.g., Poterjoy et al., 2017; Gustafsson et al., 2018; Yano et al., 2018; Bannister et al., 2020). Nevertheless, many studies have shown the benefits of using data assimilation in improving convection-allowing forecasts (e.g., Dixon et al., 2009; Brousseau et al., 2012; Tong et al., 2016; Shen et al., 2017; Tong et al., 2020; Gao et al., 2021).

The SRW v1.0.0 does not include a data assimilation capability and thus initial conditions in recent studies are purely from external models. Since the SRW will underpin the RRFS, NOAA's next generation rapidly-updated, convection-allowing ensemble forecast system, it is imperative that the data assimilation component behave as well as or better than the current operational state-of-the-art, which is the HRRR version 4. However, the first and, to our knowledge, only high resolution convection-allowing data assimilation study using the FV3 dynamical core is Tong et al. (2020), which studied the impact of the direct assimilation of radar radial velocity and reflectivity using the hybrid three dimensional ensemble–variational data assimilation (3DEnVar) and ensemble Kalman filter (EnKF) algorithms within the Gridpoint Statistical Interpolation (GSI; e.g., Wu et al., 2002; Kleist et al., 2009). Although results were for a single case study, positive impacts of assimilating radar data were found in all analyses and forecasts. Using hybrid 3DEnVar with 75 % of the ensemble background error covariance (BEC) and 25 % of the static BEC showed storm structures in the 2 h forecast comparable to when using EnKF, although EnKF outperformed 3DEnVar in the first hour forecast. Both methods, hybrid 3DEnVar and EnKF, showed higher equitable threat scores (ETS) when compared to 3DVar and pure 3DEnVar during the 4 h forecast analyzed.

Accordingly, this study seeks to describe the initial data assimilation infrastructure and performance of a prototype RRFS system. For the purpose of this paper, the prototype RRFS used is called RRFS v0.1. The focus is on extensive testing within the context of a case study to establish an understanding of baseline sensitivities, and an evaluation of various configurations and algorithms available in GSI is made in order to investigate the impact of using data assimilation on forecasts of convective storms. While single, deterministic forecasts are produced and evaluated in this study using RRFS v0.1, it should be noted that future RRFS implementations will produce convection-allowing ensemble forecasts. The 3DVar and hybrid 3DEnVar data assimilation algorithms, supersaturation removal, PBL pseudo-observations, and various weights of the ensemble BEC in the hybrid EnVar analyses are assessed. A cycling strategy is configured and its effect on the cycled analyses is evaluated. A case study that focuses on a severe convective weather event is used to demonstrate sensitivities. The RRFS_PHYv1a physics suite is adopted for the numerical simulations. Experiment results are verified using the Model Evaluation Tools (MET), which is the unified verification package that will be used by UFS applications (Brown et al., 2021). Results obtained provide developers an insight to the capabilities of RRFS developments in predicting convection, as well as suggestions for the RRFS data assimilation system framework. It is worth mentioning that despite some similarities with the work of Tong et al. (2020), in this study the focus is on the hybrid 3DEnVar method in GSI and configurations used in operational RAP and HRRR systems. For the operational RRFS, development is underway to incorporate the EnKF into the hybrid data assimilation system for its first implementation.

A brief description of each RRFS component and the corresponding workflow is presented in Sect. 2. In Sect. 3, the case study, domain, data, and experiment configurations are described. Results are presented in Sect. 4 and the summary and final remarks in Sect. 5.

## 2 Rapid Refresh Forecast System (RRFS) components

In this section, the atmospheric model, physics, data assimilation, pre-processing, and post-processing components of the RRFS v0.1 are briefly described. The workflow used to streamline all components of the system and the cycling configuration are also presented.

### 2.1 Atmospheric Model

The FV3 dynamical core was implemented in GFS replacing the spectral dynamical core for an operational upgrade in June 2019. The FV3 is a fully compressible, non-hydrostatic core featuring a Lagrangian vertical coordinate and cubed-sphere grid (Lin and Rood, 1996, 1997; Lin, 1997, 2004; Putman and Lin, 2007; Harris et al., 2020a). The Lagrangian vertical coordinate allows for a unique, straightforward representation of vertical motions directly through the relative deformation of the vertical layers. This is in contrast to the Eulerian framework presently featured in operational non-hydrostatic dynamical cores in use at the convective-scale (Skamarock et al., 2008; Janjić et al., 2001).

The FV3, originally a global model, features three types of local refinement capabilities: stretching of the global grid using the Schmidt refinement technique (Harris et al., 2016), one- and two-way nesting within the global grid (Harris and Lin, 2013), and recently a LAM capability (Black et al., 2021).The LAM capability eliminates the need to run a concurrent global model and instead relies upon lateral boundary conditions (LBCs) provided at pre-specified intervals from an external source. A more complete description of the FV3 LAM and additional justification for limited area modeling in the context of operational, convection-allowing NWP can be found in Black et al. (2021). In this study the focus is on the LAM capability, as it underpins the future RRFS and requires fewer computing resources to achieve similar forecast performance as compared to a two-way nesting method at lead times less than 24 hours (Black et al., 2021). For the FV3 LAM, initial conditions (ICs) must also be provided at least once to initiate the forecast sequence for subsequent data assimilation cycling.

### 2.2 Pre-processing

Pre-processing is performed by the utilities (UFS_UTILS) developed by NCEP's Environmental Modeling Center (EMC) (https://github.com/NOAA-EMC/UFS_UTILS) and other collaborators. UFS_UTILS can be used to generate the model grid, orography, and surface climatology (e.g. maximum snow albedo, soil, vegetation type, vegetation greenness, etc.). UFS_UTILS can also read from external models and prepare ICs and LBCs for an FV3 LAM model run.

### 2.3 Physics

The CCPP (https://dtcenter.org/community-code/common-community-physics-package-ccpp) is a collaborative effort between scientists at NOAA and the National Center for Atmospheric Research (NCAR). The goal is to assemble parameterizations developed by different groups into a common framework to be used interchangeably for numerical prediction at any scale (Heinzeller et al., 2019). Hence, the CCPP contains a set of physical schemes and a common framework that facilitates the interaction between the physics parameterizations and the dynamical core (Bernardet et al., 2020). The current common frame-

**Table 1.** RRFS_PHYv1a physics parameterizations and associated studies.

| Physical process | RRFS_PHYv1a | Associated study |
|---|---|---|
| **Shallow convection** | Mellor-Yamada-Nakanishi-Niino–eddy diffusivity-mass flux (MYNN-EDMF) | Nakanishi and Niino (2009); Olson et al. (2019) |
| **PBL/Turbulence** | Mellor-Yamada-Nakanishi-Niino–eddy diffusivity-mass flux (MYNN-EDMF) | Nakanishi and Niino (2009); Olson et al. (2019) |
| **Microphysics** | Thompson Aerosol-Aware | Thompson and Eidhammer (2014) |
| **Radiation** | GFS Rapid Radiative Transfer Model for Global Circulation Models (RRTMG) | Mlawer et al. (1997); Iacono et al. (2008) |
| **Surface layer** | GFS Surface Layer Scheme | Miyakoda and Sirutis (1986); Long (1986) |
| **Land** | GFS Noah Multi-Physics Land Surface Model | Niu et al. (2011) |
| **Gravity wave drag** | Unified Gravity Wave Physics Scheme–Version 0 | Alpert et al. (2019) |
| **Ocean** | GFS Near-Surface Sea Temperature Scheme | Li and Derber (2008); Li et al. (2015) |
| **Ozone** | GFS Ozone Photochemistry (2015) | McCormack et al. (2006, 2008) |
| **Water vapor** | GFS Stratospheric water vapor | McCormack et al. (2008) |

work was developed by the Developmental Testbed Center (DTC). A number of physics suites are available allowing great flexibility for a wide range of users. A single-column model (CCPP SCM) option has also been developed, which is available in the latest CCPP release (version 5.0, CCPPv5). The CCPPv5 supports the RRFS_PHYv1a and GFS version 16 physics suites for the SRW. The RRFS_PHYv1a suite is based on physical schemes implemented in the operational RAP, HRRR, and GFS systems and is used in all simulations in this study. Table 1 presents the RRFS_PHYv1a physics parameterizations and associated studies that describe each scheme, based on CCPP (2021).

## 2.4 Data Assimilation

GSI is a variational data assimilation system featuring 3DVar (e.g., Wu et al., 2002; Kleist et al., 2009), hybrid 3DEnVar (e.g., Wang, 2010; Wang et al., 2013; Wu et al., 2017), and hybrid 4DEnVar methods (e.g., Wang and Lei, 2014; Kleist and Ide, 2015a). It also includes an optional non-variational, complex cloud analysis capability that executes after the variational analysis as a method to specify cloud and hydrometeor variables (e.g., Hu et al., 2006a, b; Benjamin et al., 2021). GSI features the following standard control (analysis) variables: streamfunction, velocity potential, temperature, surface pressure, and normal-

ized relative humidity following Holm et al. (2002). However, the choice of control variable is flexible and one may extend or modify the standard set to include other fields, such as hydrometeors or radar reflectivity (e.g., Wang and Wang, 2017). In 3DVar and the associated hybrid variants, the static BEC is approximated through the application of a recursive filter which models the autocorrelations (Purser et al., 2003) while cross-covariances are handled in the standard context through statistical balance relationships obtained via regression (e.g., Wu et al., 2002; Parrish and Derber, 1992). The analysis is obtained by minimizing the incremental form cost function through the preconditioned conjugate gradient method (Derber and Rosati, 1989; Bathmann, 2021).

The extension of GSI from traditional 3DVar to hybrid 3DEnVar and to hybrid 4DEnVar is accomplished through the extended control variable approach (e.g., Lorenc, 2003; Wang, 2010; Kleist and Ide, 2015b, c). In this configuration, one is able to incorporate flow-dependent covariance information obtained from a complementary suite of ensemble forecasts. Typically this ensemble is obtained from a companion ensemble-based data assimilation system, such as the EnKF, however, one may use any suitably available ensemble. In fact, the regional operational data assimilation systems at NCEP have used the ensemble members from the GFS Data Assimilation System directly in the hybrid 3DEnVar framework (Wu et al., 2017). Although the use of lower-resolution global ensemble members may not be ideal for the representation of the error characteristics at finer scales, Wu et al. (2017) showed that considerable forecast improvement can be obtained even if the ensemble provided is from a different system, which is consistent with findings in other studies such as Hu et al. (2017). The present study focuses on the 3DVar and hybrid 3DEnVar frameworks and uses the global ensembles as described in Wu et al. (2017). Future work on RRFS involves the extension to a convective-scale ensemble in the EnKF, which will improve the representativeness associated with the forecast error covariance at finer scales. However, such a change is not a panacea. Aside from increased computational expense, the problem of rank deficiency of the ensemble-derived error covariance becomes more apparent with the expanded degrees of freedom associated with the finer spatial resolution. While localization helps somewhat, a computationally affordable ensemble is one that is often insufficiently sized. Therefore, future work also includes efforts to introduce multiscale data assimilation capabilities, such as scale dependent localization (e.g., Huang et al., 2021).

GSI is capable of assimilating a large suite of observations. This includes, but is not limited to, satellite radiances (e.g., Zhu et al., 2014), derived Global Navigation Satellite System Radio Occultation (GNSS-RO) observations, radar radial velocity and reflectivity (e.g., Lippi et al., 2019; Chen et al., 2021), Geostationary Lighting Mapper (GLM) lightning flash rates, web-camera derived estimates of horizontal visibility (Carley et al., 2021), and conventional observations (Hu et al., 2018). After 2014 GSI became a community system, maintained and supported by the EMC and the DTC (Shao et al., 2016). Recently, it has been added as the analysis component to improve initial conditions for the RRFS (Hu et al., 2021).

Presently, GSI is the data assimilation system used at NCEP for all operational atmospheric data assimilation applications (e.g., Wu et al., 2002; Kleist and Ide, 2015a; Hu et al., 2017). It was initially developed by the EMC (Wu et al., 2002) and implemented as the analysis component in the operational GFS in May 2007 (Kleist et al., 2009) and in the operational Rapid Refresh (RAP) in May 2012 (Benjamin et al., 2016).

## 2.5 Post-processing

UPP is used at NCEP in all operational models. A community version is currently supported and maintained by the DTC.
UPP takes native output from the model grid points/cells and creates post-processed outputs including numerous diagnostic
quantities in the same model output grid and model-native or isobaric vertical coordinate (UPP, 2021). Post-processed outputs
include diagnostic fields that are not part of the model computation and have been developed for different applications. These
include, for example, precipitation type, composite reflectivity, simulated satellite brightness temperature, updraft helicity,
storm motion, ceiling or cloud-base height, vertically integrated liquid, and lightning, among several others. More details
on the diagnostic fields developed for hourly updated NOAA weather models such as RAP and HRRR, and how they are
calculated, can be found in Benjamin et al. (2020). These products are critical for users in their forecast processes. UPP was
selected as the unified post-processing system for UFS and modifications have been made to work with FV3-based models.
Currently, it can be used in the UFS medium range weather and SRW applications.

## 2.6 Workflow

The workflow ties all RRFS components together and handles all system interdepencies. It is based on the UFS SRW appli-
cation v1.0.0 (UFS Development Team, 2021) community workflow which uses the Rocoto workflow management system
(https://github.com/christopherwharrop/rocoto/wiki/Documentation). In essence, it manages the cycling configuration, taking
into account each task dependency and specification. It oversees that tasks to generate ICs and LBCs only start if all needed
information is obtained from the previous step. It manages how the data assimilation cycle advances, i.e. by running the fore-
cast to generate the first guess and running the analysis once the first guess is completed. It handles the model execution by
supervising availability of ICs and LBCs for the specific hour, and controls that model outputs only be postprocessed if they
exist in the model run directory. It also manages crucial information on computational resource requirements to run each task.

A schematic diagram of major tasks and the general pipeline of the RRFS system is provided in Fig. 1. The task "Make Fixed
Files" generates the model grid, orography, and climatological information needed for the model execution. The tasks "Make
ICs" and "Make LBCs" read data from external models (such as GFS and HRRR), perform the necessary calculation, interpo-
lation, and conversion, and then generate appropriate ICs and LBCs for an FV3 LAM model run. The task "Run analysis" (the
gray shaded area in Fig. 1) executes the data assimilation system for an FV3 LAM run. It ingests various types of observations
and combines them with a first guess (or background) to generate a best possible atmospheric analysis for the initialization of
the FV3 LAM model integration. The first guess can be either an IC from an external model (after the task "Make ICs") or a
short term forecast (1 − 6 h forecast, configurable to users) from a previous FV3 LAM model run. The first scenario is referred
to as a "cold start" (the blue box in Fig. 1) while the latter is called a "warm start" (the red box in Fig. 1). In practice, for an
FV3 LAM "warm start," the first guess comes from "restart" forecast files generated by the FV3 LAM model. The task "Run
model" is to run the FV3 LAM model with ICs and LBCs prepared from the previous steps. It is worth mentioning that besides
the "cold start" and "warm start," an FV3 LAM model run can also start from an IC made directly from an external model

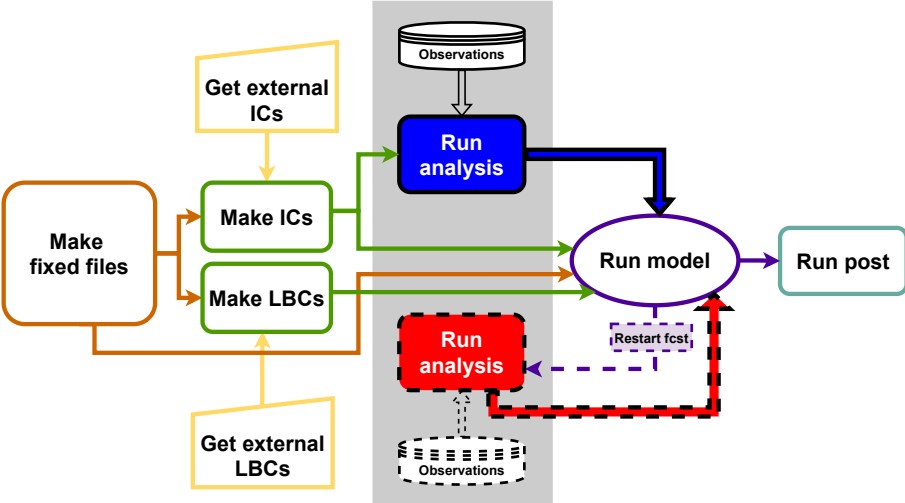

**Figure 1.** Schematic diagram of the RRFS tasks and workflow.

without the data assimilation step. This is also referred as a "cold start." The task "Run post" is to post-process the FV3 LAM forecasts and generate all target model fields for downstream plotting and/or examination.

## 2.7 Cycling configuration

The cycling configuration of the RRFS v0.1 is similar to the one used in RAP, i.e. cold starts are performed every 12 hours and warm starts are performed at all other cycles using the 1 h forecast from the previous cycle as background for the analysis. RAP performs hourly-updated continuous cycles with cold starts at 09:00 UTC and 21:00 UTC using the 1 h forecast from cycles initialized at 08:00 UTC and 20:00 UTC in 6 h parallel hourly spin-up cycles. The parallel spin-up cycles are cold started from GFS atmosphere analyses and RAP surface fields at 03:00 UTC and 15:00 UTC. Cold starts in RAP introduce the atmospheric conditions while RAP land surface fields are fully cycled in the continuous cycle (Benjamin et al., 2016; Hu et al., 2017). Periodic updates of the large scale atmospheric conditions are needed in regional modeling systems in order to account for corrections made by global observations over land and ocean and to avoid model drift from those conditions (Benjamin et al., 2016). At the time of execution of this research not many RAP functionalities were available for use in the RRFS data assimilation framework, therefore a simplified configuration with partial cycling is used. Development currently underway includes establishing a partial cycling capability for the inaugural operational implementation, RRFS version 1, with subsequent plans to consider a fully cycled version in later implementations leveraging recent advances discussed in Schwartz et al. (2022). Figure 2 illustrates the RRFS cycling configuration from cycles initialized between 00:00 UTC 12:00 UTC. In each cycle, an 18 h free forecast is launched following the analysis, with hourly outputs. A cold start is performed at 00:00 UTC and 12:00 UTC and warm starts between 01:00 UTC to 11:00 UTC using the FV3 LAM 1 h forecast from the previous cycle as background for the analysis.

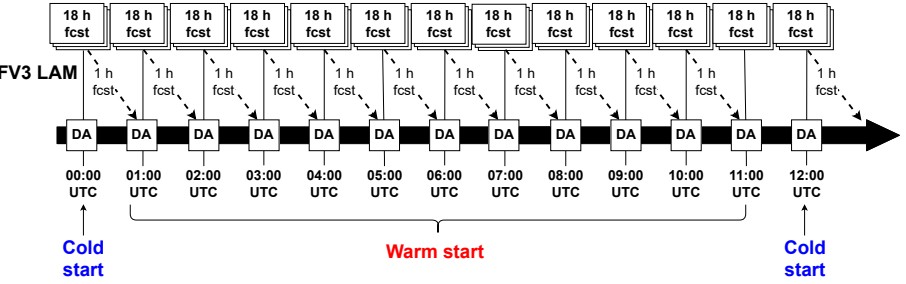

**Figure 2.** RRFS cycling configuration diagram.

## 3 Methods

In order to achieve skillful forecasts comparable to the current operational convection-allowing suite, each component of the RRFS needs to be exhaustively tested to determine the best configuration. This study focuses on the initial configuration of the data assimilation framework. In this section, the case study, general setup of the experiments, description of the experiments conducted, and verification methodology are presented.

### 3.1 Case overview

A line of convective storms developed over northeastern Oklahoma ahead of a southward moving cold front during the afternoon of 4 May 2020. At 18:00 UTC on 4 May 2020, a surface low pressure was observed across western Oklahoma with a dry line extended over western Texas, favoring an environment with low-level convergence, high temperatures, and humidity over these areas. Between 19:00 UTC and 20:00 UTC, high values of mixed layer convective available potential energy (ML-CAPE) (3694 J kg$^{-1}$) and effective bulk shear (48 kt for the surface to 3 km layers and 36 kt for the surface to 6 km shear)

were observed over northeastern Oklahoma. The instability parameters are based on the observed sounding at 19:00 UTC over Norman, OKlahoma (KOUN). This environment provided favorable conditions for severe convective storms with potential for strong updrafts and development of supercells (e.g., Weisman and Klemp, 1982; McCaul and Weisman, 2001). At 20:00 UTC, convective cells were first seen in the radar reflectivity observations over that region (Fig. 3a), and at around 22:00 UTC (Fig. 3c) a line of storms extended across central Oklahoma along the pre-frontal wind shift. The system evolved while slowly

moving southeastward. A supercell developed over far southwestern Missouri at 00:00 UTC on 5 May (Fig. 3e), producing hail of 1.25 and 1.5 inches in diameter according to hail reports from the Storm Prediction Center (SPC). Clusters of severe storms developed across south-central Oklahoma along the intersection of the cold front with the dry line. The convection associated with the squall line evolution resulted in several instances of large hail and high wind, mostly over northeastern and south-central Oklahoma, southeastern Kansas, southwestern Missouri, and northwestern Arkansas.

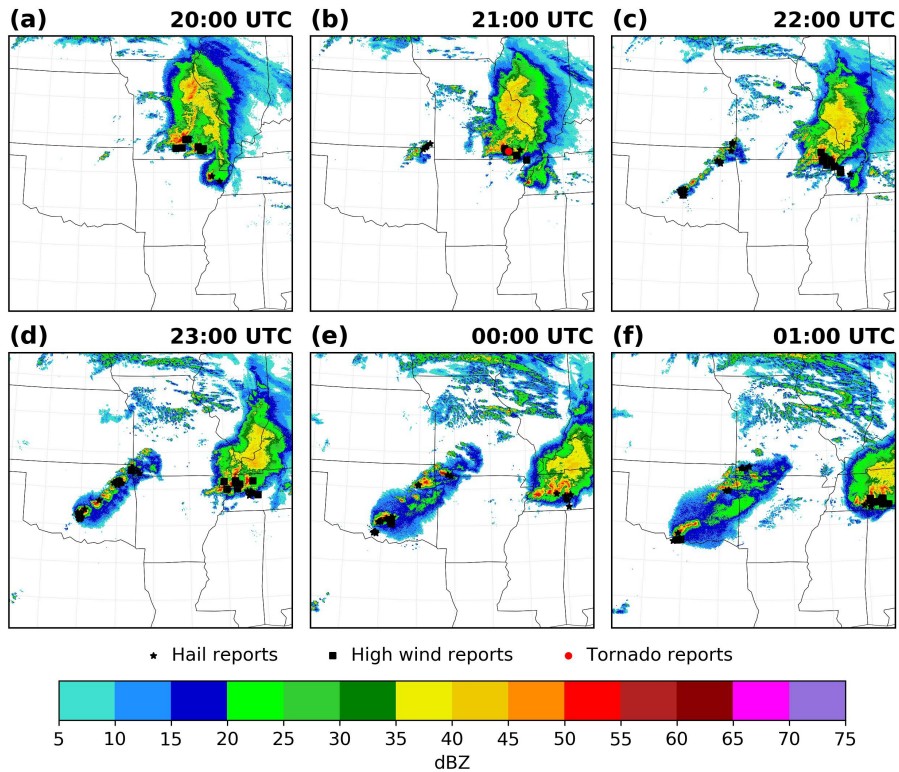

**Figure 3.** Hourly Multi-Radar Multi-Sensor (MRMS) composite reflectivity and hourly hail (black stars), high wind (black squares), and tornado (red circles) reports from the SPC, from 20:00 UTC on 4 May 2020 through 01:00 UTC on 5 May 2020.

### 265 **3.2 Setup of experiments**

For the simulation of this case, a domain is configured consisting of $460 \times 460$ grid cells centered on Fort Smith, Arkansas with a 3 km horizontal grid-spacing and 65 vertical layers. All simulations start at 00:00 UTC on 4 May 2020 and run hourly cycles until 06:00 UTC on 5 May 2020. Hourly 3 km HRRR analyses and forecasts are used to generate the ICs and LBCs for the FV3 LAM. The observation data assimilated in each experiment are the same as those used in the operational RAP

system (Hu et al., 2017) and includes upper air observations from rawindsondes, dropsondes, pilot balloons, aircraft, and wind profilers; surface data from synoptic stations, METeorological Aerodrome Reports (METAR), and the Mesoscale Network (MESONET); radar radial velocity and the vertical azimuth display derived from radar radial velocity; Atmospheric Motion Vectors (AMV) wind derived from satellite observations; and the Global Positioning System (GPS) Integrated Precipitable Water (GPS-IPW). The time window used is 1 hour, allowing for observations to be assimilated within 30 minutes before to

30 minutes after the central analysis time.

Experiments are conducted testing the GSI 3DVar and 3DEnVar systems. For the hybrid 3DEnVar analysis, the Global Data Assimilation System (GDAS) 80 member ensemble forecasts (9 h forecasts) are used to provide the ensemble BEC (e.g.,

Wu et al., 2017). These forecasts have a horizontal resolution of approximately 25 km and are available four times per day, therefore the same 9 h GDAS ensemble forecasts are used for the 2 hours before and 3 hours after its valid hour. For example, the 9 h GDAS ensemble forecasts initialized at 00:00 UTC (valid at 09:00 UTC) are used for the cycles from 07:00 UTC to 12:00 UTC. Similarly, the 9 h forecast GDAS ensemble initialized at 06:00 UTC (valid at 15:00 UTC) is used for the cycles from 13:00 UTC to 18:00 UTC. This follows the same strategy in the RAP system (Hu et al., 2017). As shown in Hu et al. (2017), using off-time global and fixed ensemble-based BEC still produces better results than only using the static BEC. In all experiments with data assimilation, two outer loops with 50 iterations per loop are performed to minimize the cost function and find each analysis. In each outer loop a re-linearization is performed (e.g., Kleist et al., 2009). The increment is zero for the first outer loop while for the second it is updated with the solution found after the 50 iterations of the first outer loop. The spatial resolution of the analysis is 3 km, as in the forecast model.

### 3.3 Sensitivity experiments

GSI provides many functionalities and parameters, enabling users to make the best data assimilation configurations for different applications. A series of experiments are designed to examine the impact of different configurations on the analyses and forecasts. Some RAP configurations are tested in the experiments following Hu et al. (2017). An experiment with no data assimilation is provided, acting as the baseline for all other experiments. This baseline experiment is called NoDA and uses the same cycling configuration as experiments with data assimilation, in terms of the cold and warm start ICs. The 3 km ICs from the HRRR are consistent with the 3 km grid-spacing of the RRFS, such that fine scale features found in the HRRR are present in the RRFS ICs. Table 2 lists all experiments in this research.

In order to examine how different weights of the ensemble BEC affect the results and what would be the best choice for the RRFS analysis, experiments with different ensemble weights are conducted. Only results from three experiments are presented here, i.e. 3DVar, 100EnBEC, and 75EnBEC. The experiment with 3DVar does not include any ensemble BEC part, 100EnBEC uses pure ensemble BEC and does not include the static part, and 75EnBEC uses a combination of 75 % ensemble BEC and 25 % static BEC. The static BEC for 3DVar is the same as currently used in RAP and HRRR (Benjamin et al., 2016).

Ensemble localization length scales play an important role in ensemble-based data assimilation algorithms, such as hybrid EnVar analyses (e.g., Campbell et al., 2010), as an effective way to mitigate sampling errors due to the relatively small ensemble size available for hybrid EnVar and ensemble analyses (Houtekamer and Mitchell, 2001; Hamill et al., 2001), especially at convective scales (e.g., Gustafsson et al., 2018). At this stage, it is important to determine how large the localization radius needs to be for RRFS analyses. Therefore, experiment VLOC was designed to examine the vertical localization radius that yields more realistic forecasts in RRFS. A separate study is underway in which the optimal horizontal localization for RRFS is also investigated and therefore it is not examined here. The localization function in GSI is implemented as a single application of an isotropic recursive filter (Purser et al., 2003) and the radius is specified as a Gaussian half-width, either in scale height ($\ln p$) or in terms of number of vertical layers. In this study, the radius is specified in terms of the number of layers. In VLOC, the vertical ensemble localization radius is changed from 3 vertical layers for the whole atmosphere (used in all other experiments) to a height-dependent localization setting: 1 vertical layer in the lowest 10 model layers and 3 layers for other model layers. A

comparison experiment (not shown) was conducted reducing the vertical localization to 2 layers in the first 10 model layers, but results showed neutral impacts over VLOC.

The operational RAP system has developed a PBL pseudo-observation function in order to obtain a better representation of the PBL in the analysis. This function was initially developed to further leverage the information provided by METAR observations, extending their representativeness through the PBL depth in the Rapid Update Cycle (RUC) analyses. Improvements in the temperature, dew point, and CAPE forecasts were found when spreading the innovations from temperature, moisture, and wind in the layers above the surface and below the top of the PBL (Benjamin et al., 2004). Smith et al. (2007) also found a positive impact in the 3 h forecast of CAPE by using the PBL pseudo observations, and the impact was greatly increased when additionally assimilating GPS-IPW. Benjamin et al. (2010) found higher positive impact during the summer, when the PBL is deeper. This function has been used operationally since RAP version 3 (Benjamin et al., 2016), and therefore, it needs to be tested and tuned for its potential use in RRFS analyses. To test whether and how this function works for the RRFS v0.1, experiment PSEUDO is designed and results are presented in Sect. 4.3.

The study of Tong et al. (2020) showed that regardless of the method used, the storm coverage was overestimated and reflectivity values were much higher than observed, which is likely linked to the physics suite used. However, it is also well known that nonphysical solutions (nonrealistic updraft/downdraft, negative humidity, supersaturation, etc.) can arise as a result from the data assimilation procedure (e.g., Janjić et al., 2014; Tong et al., 2016). In this study, experiment CLIPSAT is conducted to analyze how the supersaturation removal procedure available in GSI affects the storm forecasts of RRFS. This function constitutes a simple adjustment in the background supersaturation during the cost function minimization. More details on this function are presented in Sect. 4.4.

Experiments VLOC, PSEUDO, and CLIPSAT are performed using the hybrid 3DEnVar algorithm with 75 % of the ensemble BEC and compared against 75EnBEC results, due to the good results obtained for experiment 75EnBEC (see Sect. 4.2) and the consideration that RAP uses 75 % of the ensemble BEC operationally (Hu et al., 2017).

## 3.4 Forecast verification

MET version 9.0 (Jensen et al., 2020) is used for forecast verification. MET was developed at the DTC and has been widely used by the NWP community. Upper air and surface observations are used to verify the vertical profiles of temperature, specific humidity, and wind, as well as 2 m temperature and 2 m dew point, respectively. For upper air observations, the verification time window is 1 hour and 30 minutes before to 1 hour and 30 minutes after, while for surface observations, it is 15 minutes before to 15 minutes after the central time. The root mean square error (RMSE) and bias are computed, displayed with 95 % confidence intervals that are derived using a bootstrap resampling technique of 1000 replications with replacement at each forecast lead hour in every cycle, and with bias-corrected percentiles (e.g., Wilks, 2006; Gilleland et al., 2018). Upper air statistics are further analyzed at 00:00 UTC and 12:00 UTC valid times.

Precipitation forecasts are verified against the hourly Stage IV precipitation product (Lin and Mitchell, 2005) in terms of the ETS and frequency bias (FBIAS) for different thresholds, but only $>0.01$ inches $\mathrm{h}^{-1}$ (0.254 mm $\mathrm{h}^{-1}$) for lighter precipitation and $>0.25$ inches $\mathrm{h}^{-1}$ (6.35 mm $\mathrm{h}^{-1}$) for heavier precipitation are presented here. The grid-to-grid approach in MET is used.

**Table 2.** List of experiments presented in this study.

| Experiments | Background error covariance weights | Supersaturation removal | PBL pseudo-observations | Vertical ensemble localization scale |
|---|---|---|---|---|
| NoDA | No data assimilation | | | |
| 3DVar | **0 % ensemble**<br>100 % static | false | false | 3 layers |
| 100EnBEC | **100 % ensemble**<br>0 % static | false | false | 3 layers |
| 75EnBEC | **75 % ensemble**<br>25 % static | false | false | 3 layers |
| CLIPSAT | **75 % ensemble**<br>25 % static | **true** | false | 3 layers |
| PSEUDO | **75 % ensemble**<br>25 % static | false | **true** | 3 layers |
| VLOC | **75 % ensemble**<br>25 % static | false | false | **1 layer in first 10 layers and 3 layers above** |

Hourly MRMS composite reflectivity mosaics (optimal method) observations (Zhang et al., 2016) are used to verify the composite reflectivity forecasts using the Method for Object-Based Diagnostic Evaluation (MODE) in MET. In order to quantitatively identify the experiment configuration that yielded better forecasts, the median of maximum interest (MMI (F+O)) (Davis et al., 2009) is analyzed. This metric results from the median between the maximum interest from each observed object with all predicted objects (MIF), and the maximum interest from each predicted object with all observed objects (MIO). It takes into account all attributes used in the total interest calculation, summarizing them into a single value. The forecast in greatest agreement with the observations will give MMI (F+O) values closer to one. Otherwise, the values will be closer to zero.

## 4 Results and discussions

### 4.1 Examination of Analyses

Observation availability and coverage play an important role in the data assimilation process. Therefore, how many and what type of observations are available for this squall line case are examined. Figure 4 shows the spatial distribution of assimilated temperature observations at the 19:00 UTC cycle on 4 May 2020 for experiment 3DVar (other cycles and experiments have similar distributions and are not shown here). The analysis residuals (OmA) are also depicted in Fig. 4 using red and blue color depth for positive and negative values, respectively. In this analysis, assimilated temperature observations include those from

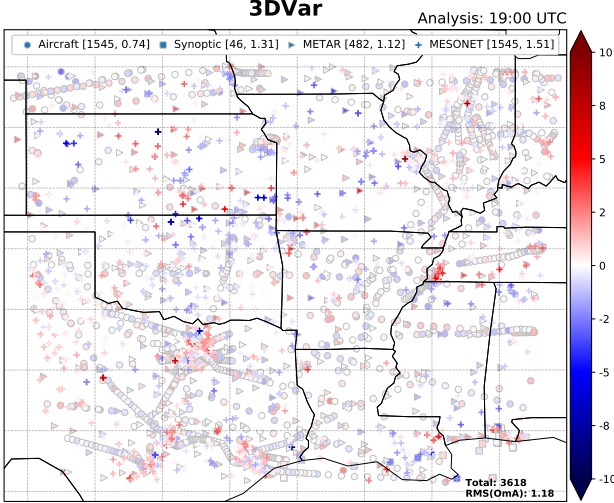

**Figure 4.** Spatial distribution of temperature observations and analysis residuals (OmA) for the analysis at 19:00 UTC on 4 May 2020 from experiment 3DVar. The color scale to the right indicates the magnitude of analysis residuals. The legend of observation type markers is shown at the top along with brackets listing associated counts and Root Mean Square (RMS) error for the OmA.

aircraft, surface marine synoptic stations, METAR, and MESONET observations. There are a total of 3307 observations, which are well distributed across the limited model domain. Among these observations, 1545 are from aircraft, which concentrate around a few major airports as flights descend or ascend, and spread along flight paths. Moreover, a substantial amount of MESONET surface observations are also assimilated. There are far fewer METAR observations, but they are distributed evenly in the domain. A very limited number of surface marine synoptic stations are found near the coast on the Gulf of Mexico. The analysis residuals for temperature are generally small for aircraft and METAR observations, mostly less than $\pm 1°$ K in magnitude, while some MESONET observations have large analysis residuals. As pointed out in Morris et al. (2020), while some MESONET stations are well maintained, the majority do not meet siting standards and maintenance protocols and therefore are assigned a higher observation error via a station blacklist. As expected, larger residuals are found from these observations when compared to other observation networks.

In order to check how results of the RRFS analysis behave at different cycles and whether it executes correctly, Fig. 5 presents time series of the RMS error and bias for the backgrounds (1 h forecasts) as well as the analyses, verified against temperature observations (including all surface and upper air data as mentioned in Sect. 3.2). Results presented are for experiments 75En-BEC (Fig. 5a) and 3DVar (Fig. 5b). Verification is conducted by utilizing the observation innovations (OmB) and the analysis residuals (OmA) generated by GSI for assimilated observations. Based on these OmB and OmA data, the RMS error and bias for the background and analysis are computed. It can be seen from Fig. 5 that the analyses have smaller RMS errors and biases compared to the background in both experiments. This means the analyses fit the observations more closely, though owing to observation error not perfectly, which is expected from a correctly executed data assimilation procedure. There is a noticeable

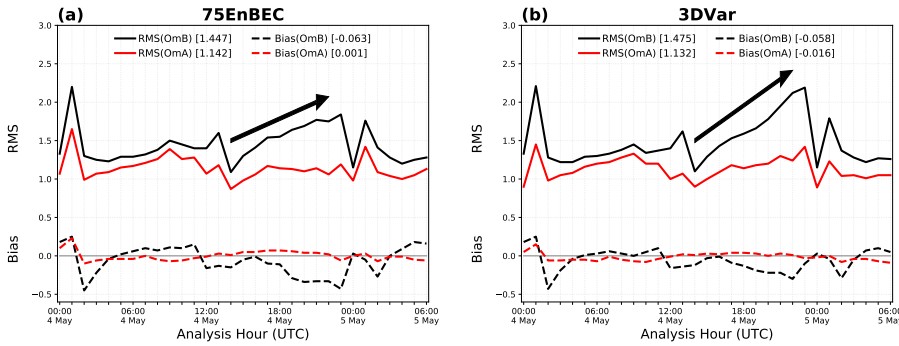

**Figure 5.** RMS, bias, and count of the temperature background (OmB) and analysis (OmA) against all observation types for analyses in all cycles performed for experiments (a) 75EnBEC and (b) 3DVar. Black arrows highlight the time period from 14:00 UTC to 23:00 UTC.

jump in the RMS error values of the OmB from 00:00 UTC (12:00 UTC) to 01:00 UTC (13:00 UTC) on 4 May 2020. This is because 00:00 UTC and 12:00 UTC are cold started from HRRR analyses. On the contrary, at 01:00 UTC (13:00 UTC) on 4 May, the background used is from the FV3 LAM 1 h forecast. Therefore, forecasts used to initialize cycles at 01:00 UTC and 13:00 UTC undergo a spin-up process. The FV3 LAM 1 h forecasts are still in this spin-up process and hence yield larger RMS errors. In Fig. 5a, the background RMS error increases steadily from 14:00 UTC to 23:00 UTC, compared to the relatively gentle increase between 02:00 UTC to 11:00 UTC on 4 May. This may be due to the fact that there is active convection during the afternoon hours and, hence, it is harder to obtain good forecast skill. Figure 5b has a much larger increase in the RMS error of OmB than that in Fig. 5a during the same time period from 14:00 UTC to 23:00 UTC, indicating that 75EnBEC performs better than 3DVar. Results from 100EnBEC are similar to 75EnBEC and are not shown here.

## 4.2 The impact of hybrid ensemble weights and ensemble localization radius

### 4.2.1 The impact of hybrid ensemble weights

The hybrid EnVar data assimilation method is now widely used by NWP centers (e.g., Bannister, 2017; Gustafsson et al., 2018) and the research community. It combines the static and ensemble BEC, taking advantages from both the variational method and the EnKF method. It is robust, allows the use of flow-dependent BEC, avoids the development and maintenance of a tangent linear and adjoint model, and thus has gained mainstream practice. In this hourly updated RRFS system, the hybrid 3DEnVar method is tested. One of the major concerns is to how to obtain the optimal weight for the ensemble BEC in the hybrid 3DEnVar analysis. A series of weighting sensitivity experiments were conducted in order to find the best option for this study.

Figure 6 shows the specific humidity and temperature analysis increments for the 19:00 UTC cycle on 4 May 2020 for experiments 100EnBEC, 75EnBEC, and 3DVar. The analysis conducted at 19:00 UTC is during a cycling period using warm starts and is close in time to the initiation of convection in the afternoon hours. Forecasts initialized by this analysis cover

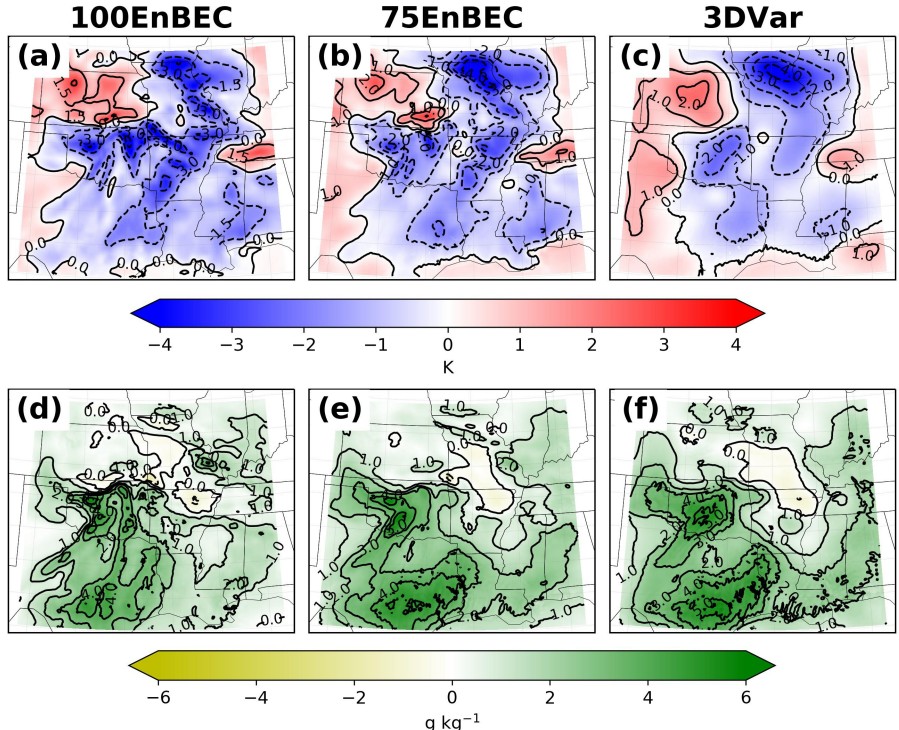

**Figure 6.** Analysis increment for temperature (K) (a, b, and c) and specific humidity ($\mathrm{g\,kg^{-1}}$) (d, e, and f) at the first model hybrid level above the surface for 19:00 UTC on 4 May 2020, using 100 % ensemble BEC (a and d), 75 % ensemble BEC (b and e), and 3DVar (c and f).

the squall line evolution from its initiation to decay stages. Therefore, this cycle is selected to show the analysis increments
and storm forecasts in the following sections. The analysis increments from experiment 3DVar (Fig. 6c and f) are smoother as compared to those from 75EnBEC (Fig. 6b and e), which exhibits some flow-dependent features. As it goes into pure ensemble BEC (Fig. 6c and d), more flow-dependent increments are obtained.

Figure 7 shows the 2, 4, and 6 h forecasts of composite reflectivity from the 19:00 UTC cycle on 4 May 2020, with 5 $\mathrm{dBZ}$ (solid lines) and 35 $\mathrm{dBZ}$ (dash lines) reflectivity observation contours overlaid for experiments 100EnBEC, 75EnBEC, 3DVar,
and NoDA. The regridding tool in MET is used to interpolate the MRMS composite reflectivity observations to the same grid as the model forecasts. Additionally, MMI (F+O) results for reflectivity values larger than 35 $\mathrm{dBZ}$ for each experiment are shown in the lower right corner of each panel. All experiments predict the general evolution of the squall line, from the initial stage to maturity, with overforecasting of high reflectivity values and underforecasting of spatial coverage. At the 2 h forecast, the experiments capture the convective initiation around northeastern Oklahoma, but the extent and intensity of the
cells are overpredicted (Fig. 7a, d, g, and j). The initial cells are represented and located more accurately in the experiments with data assimilation, especially 75EnBEC with a MMI (F+O) value of 0.540 (Fig. 7d). In the 4 h forecast, the squall line enters its mature stage and a line of storms are ranged from southwest Missouri to central Oklahoma (Fig. 7b, e, h, and k).

Every experiment predicts a squall line, but there is substantial location and coverage error in the NoDA experiment. 3DVar improves a little over NoDA, but due to the difference in the coverage predicted, a decrease in the MMI (F+O) value from 0.632 to 0.556 is observed. 75EnBEC does well to predict the squall line at the correct location with the larger MMI (F+O) value of 0.698, although the storm near the southwest tip of the observed squall line is still missing, as it is in all other experiments (Fig. 7e). 100EnBEC overproduces the convection associated with the squall line, but still improves over 3DVar and NoDA at this forecast hour. In the 6 h forecast, the squall line moves eastward and covers from southern Missouri and northwestern Arkansas to southeastern Oklahoma. At this time, 3DVar again performs better than NoDA with very close MMI (F+O) results, and 75EnBEC still makes the best forecast among all experiments (Fig. 7c, f, i, and l). However, in terms of the MMI (F+O) values, 75EnBEC shows a slight degradation for the forecast of reflectivity values larger than 35 dBZ, and 100EnBEC shows the best MMI (F+O) value of 0.555. Overall, data assimilation introduces evident, positive impacts to the storm forecasts in terms of the squall line location, orientation and coverage, though different assimilation strategies yield different impacts. The improvement from 3DVar is somewhat limited while hybrid 3DEnVar is seen to perform much better. Among the experiments, the 75% ensemble BEC gives the best overall forecasts.

Vertical profiles of RMSE and bias with 95 % confidence intervals for the 2 h forecast of temperature, specific humidity, and wind at 00:00 UTC and 12:00 UTC valid hours (from cycles initialized at 22:00 UTC and 10:00 UTC on 4 May, respectively) are shown in Fig. 8. The confidence intervals help to highlight where the differences between the experiments are statistically significant. Experiments show a consistent warm bias at both 00:00 UTC and 12:00 UTC in most vertical levels (Fig. 8a and d). A cold temperature bias is present in the layers between 850 and 650 hPa at 00:00 UTC and at 1000 hPa and 150 hPa at 12:00 UTC in all experiments. Experiment 75EnBEC has smaller temperature RMSE values between 400 and 250 hPa at 00:00 UTC and between 550 and 400 hPa at 12:00 UTC. The improvements for the temperature bias at 00:00 UTC are statistically significant between 500 and 400 hPa. Experiment 100EnBEC shows smaller RMSE at 850 hPa at both valid hours. All experiments with data assimilation have smaller temperature RMSE and bias below 850 hPa for 00:00 UTC which are statistically significant as shown by the confidence intervals, indicating the positive impact from data assimilation in the lower atmosphere. The impact of the analysis on the 2 h temperature forecast valid at 12:00 UTC is less clear. Similarly, the specific humidity forecasts show improved RMSE and bias from data assimilation with statistically significant differences below 900 hPa at 00:00 UTC and between 750 and 500 hPa at 12:00 UTC in the bias results (Fig. 8b and e). The 2 h forecast of wind profiles has a positive bias in the lower levels at both valid hours, but mostly negative above 850 hPa (Fig. 8c and f). The positive impact of using data assimilation is clearly observed in the winds close to the surface for levels below 950 hPa, where there are statistically significant differences in bias between the experiments with data assimilation and NoDA. The wind RMSE results do not clearly indicate which experiment is best, but in general 100EnBEC shows the lowest values when considering all vertical levels. These results may indicate that the static BEC matrix used may not be optimal for RRFS v0.1 and efforts are underway in order to obtain a better BEC matrix. Moreover, an online estimation approach may be explored for the specification of the hybrid weighting parameter, such as the method proposed by Azevedo et al. (2020) in which a geographically varying weighting factor alpha is defined and the ensemble spread is used for the assignment of the weights.

Figure 9 presents the RMSE and bias for the 2 h forecast of 2 m temperature (Fig. 9a and c) and 2 m dew point temperature (Fig. 9b and d) for experiments 100EnBEC, 75EnBEC, 3DVar, and NoDA. 2 m temperature and 2 m dew point RMSE are evidently larger between cycles initialized at 16:00 UTC and 23:00 UTC in all experiments. This may be related to the initiation and development of convection in many areas of the domain. During this period, all data assimilation experiments have smaller 2 m temperature and 2 m dew point RMS errors compared to the NoDA experiment, demonstrating the positive impact from data assimilation. Further, experiments 75EnBEC and 3DVar perform better than 100EnBEC between cycles initialized at 16:00 UTC and 20:00 UTC (18:00 UTC and 22:00 UTC valid hour) (Fig. 9a). Among them, 75EnBEC produces the smallest 2 m temperature and 2 m dew point RMSE. From 16:00 UTC to 23:00 UTC valid hour, the 2 h forecasts from all experiments show a warm and dry bias. Data assimilation experiments helped to reduce this warm and dry bias to some extent.

To summarize, experiment 75EnBEC performs reasonably better among all experiments discussed in this section. It gives the smallest 2 m temperature and 2 m dew point RMSE during the afternoon storm hours and a better representation of the storm in all forecasts lengths. Therefore, all subsequent experiments use the 75 % ensemble BEC.

### 4.2.2 The impact of vertical ensemble localization radius

Ensemble-based systems need a large number of ensemble forecasts in order to estimate a full rank covariance matrix. However, this is computationally impractical for operational and research activities. The ensemble-based covariances can be very noisy when using a small ensemble size which results in inaccurate analyses (e.g., Hamill et al., 2001; Gustafsson et al., 2018). The vertical and horizontal localization scales determine how the ensemble covariance varies with distance (Buehner, 2005). Gustafsson et al. (2018) pointed out that the localization needs to be large enough to not disrupt the large scale balance but small enough to represent fluctuations at the convective scale. Thus, unlike at global scales, the operational RAP and HRRR systems use a horizontal localization radius of 110 km in combination with a vertical localization radius of 3 layers, which gives optimal forecast skill in RAP applications (Hu et al., 2017). In addition, Hu et al. (2017) tested a vertical localization radius of 9 layers, but using this larger localization radius degraded the forecast when compared to 3 layers. Knowing the expected results for a relatively larger vertical localization value using an 80 member ensemble, this study looks at the impact of reducing the vertical localization radius from 3 grid points to 1 in the lowest 10 vertical model levels (experiment VLOC). This reduction is adopted to capture finer vertical features of the low atmosphere from observations close to the surface and below the PBL.

Figure 10 presents the RMSE and bias with 95 % confidence intervals for vertical profiles of the 2 h forecast of temperature, specific humidity, and wind valid at 00:00 UTC and 12:00 UTC for experiments VLOC and 75EnBEC. For the temperature forecasts, VLOC has a lower RMSE between 800 and 550 hPa and smaller bias in the lower atmosphere between 1000 to 900 hPa and 800 to 700 hPa during the late afternoon (valid hour 00:00 UTC) (Fig. 10a). At valid hour 12:00 UTC, VLOC gives a lower RMSE between 950 and 900 hPa and 350 and 300 hPa, and lower bias in the upper atmosphere between 450 and 250 hPa (Fig. 10d). For specific humidity, the RMSE and bias are improved at all levels above 650 hPa at 00:00 UTC with VLOC. However, a degradation is observed in the RMSE in the lower levels below 700 hPa. Degradation is also seen in the bias between 950 and 800 hPa (Fig. 10b). At valid hour 12:00 UTC, not much improvement is shown in either the RMSE or

bias from VLOC (Fig. 10e). Most of the differences between these experiments are not statistically significant as indicated by the confidence intervals. Meanwhile, a general positive impact is observed in the RMSE and bias for the winds above 650 hPa, being statistically significant at 300 hPa for both, and in the RMSE and bias results at valid hour 00:00 UTC. Negative impact is found in lower levels (Fig. 10c). At 12:00 UTC valid hour, slight improvements are shown for VLOC in the RMSE between 650 and 500 hPa and at 400 hPa, and in the bias at 550 hPa and 350 hPa, but the differences are not statistically significant (Fig. 10f).

The change in vertical localization slightly improves the extent and intensity of convection over northeastern Oklahoma in the 2 h forecast, however MMI (F+O) values indicate that the experiment 75EnBEC is still more skillful at representing reflectivity larger than 35 dBZ with a decrease from 0.540 in 75EnBEC to 0.528 in VLOC (Fig. 11a). An underforecast of the convection over central and eastern Oklahoma is observed in VLOC in the 4 h forecast with a smaller MMI (F+O) value of 0.587(Fig. 11b) and an overforecast over north-central Arkansas and south-central Missouri is observed in the 6 h forecast with a slight improvement in the MMI (F+O) value from 0.544 in 75EnBEC to 0.563 in VLOC (Fig. 11c). While reducing the vertical localization scale did produce small improvements at some vertical levels and larger forecast lengths, degradation dominated the overall signature, indicating this variation of localization scale produces overall less skillful storm forecasts. The analysis cycling technique and multivariate relationships in the BEC spread the impact of the observations throughout different levels and locations, which could have led to the slight positive impact above 650 hPa instead of the lower atmosphere where the modification in the vertical localization is made. It suggests a vertical ensemble localization radius of 3 layers is already a good choice if not the best.

### 4.3   The impact of PBL pseudo-observations

The impact of adding PBL pseudo-observations to the analysis based on surface temperature and moisture observations is evaluated in experiment PSEUDO. This function first identifies the PBL height using the background (FV3 LAM 1 h forecast). Then, using METAR observations, it computes the 2 m temperature and 2 m moisture observation innovations (OmB) such that they are inserted at multiple vertical levels, from the surface to the level corresponding to 75 % of the PBL height and spaced every 20 hPa (Benjamin et al., 2016). This technique works as if additional PBL observations are available at those levels and thus more observation innovations can be computed. Therefore, they are called "PBL pseudo-observations". This function is tested with the PBL pseudo-observation configuration used in the operational RAP system.

The 2, 4, and 6 h composite reflectivity forecasts from experiments PSEUDO and 75EnBEC are presented in Fig. 12. PSEUDO clearly predicted more convection than 75EnBEC in the 2 h forecast, with a smaller MMI (F+O) value of 0.533 in comparison to 0.540 in the experiment 75EnBEC (Fig. 12a). However, noticeable improvements in the coverage and positioning of the storm are found in 4 and 6 h forecasts, with a corresponding increase in the MMI (F+O) values when compared to 75EnBEC (Fig. 12b and c). Especially at 4 h, the representation of the squall line over Oklahoma is greatly improved after adding PBL pseudo-observations with a better coverage of the squall line, although an increase in the intensity of the convective cores is also noted (Fig. 12b). Spurious convection also appeared over northwest Oklahoma and Texas in the 4 h forecast and

over Texas in the 6 h forecast. These results indicate the potential of using PBL pseudo-observations in RRFS to improve the representation of convection.

The RMSE and bias vertical profiles for the 2 h forecast of temperature, specific humidity, and wind against sounding observations at the 00:00 UTC and 12:00 UTC valid hours are presented in Fig. 13. The use of PBL pseudo-observations gives subtle positive impacts at both valid hours and most vertical levels for the RMSE and bias of temperature and specific humidity (Fig. 13a, b, d, and e). Improvements in the RMSE and bias of temperature are observed below 900 hPa at valid hour 00:00 UTC. The positive impact in the bias extends to 800 hPa, indicating the better representation of the lower atmosphere in the experiment PSEUDO (Fig. 13a). A slight degradation is observed in the middle levels at the same valid hour. For wind, the RMSE shows more promising results with a positive and statistically significant impact between 500 and 550 hPa at 00:00 UTC and 12:00 UTC valid hours. This positive impact is also significant at 300 hPa in the RMSE and bias results at 00:00 UTC (Fig. 13c). At 12:00 UTC, the bias shows more subtle improvements in 750 hPa and 300 hPa (Fig. 13f).

Similar to the upper air verification, the RMSE and bias of 2 m temperature and 2 m dew point temperature for the 2 h forecast in PSEUDO show overall neutral impact. A degradation in the RMSE of 2 m temperature is observed between cycles initialized at 21:00 UTC and 00:00 UTC (Fig. 14a) and in the bias between cycles initialized at 17:00 UTC and 21:00 UTC. A subtle improvement is seen in the bias of 2 m temperature between cycles initialized at 21:00 UTC and 23:00 UTC. Slight improvements are observed in the RMSE and bias of 2 m dew point temperature between cycles initialized at 19:00 UTC and 23:00 UTC. Adding PBL pseudo-observations helps to mitigate near surface dry bias during afternoon hours, makes upper air forecasts better in some levels of the middle and upper atmosphere, and clearly improves the storm forecast in the 4 h and 6 h forecasts. Nevertheless, more tuning and testing of this function are needed before applying this technique in the RRFS.

## 4.4   The impact of supersaturation removal

GSI has a function to remove supersaturation in the background by capping specific humidity to its saturation value in each outer loop during the minimization of the cost function, as calculated using the background fields (CIMSS, 2014). Figure 15 shows the difference in the specific humidity ($g\,kg^{-1}$) analyses between the 75EnBEC analysis and the 75EnBEC analysis with the supersaturation clipping function activated (75EnBEC vs. CLIPSAT) for the 19:00 UTC cycle on 4 May 2020. Since more moisture is present in the lower atmosphere, model hybrid level 15 (located in the lower atmosphere at around 850 hPa) is selected to show this result. Positive (negative) differences in Fig. 15 indicate that more (less) specific humidity is found in the 75EnBEC analysis than in CLIPSAT. The figure suggests that supersaturation is removed in the CLIPSAT analysis mostly over southwestern and northwestern Missouri, southeastern Kansas, northern Arkansas and Oklahoma. It is worth mentioning that the computational run time of the analyses in CLIPSAT is quite similar to 75EnBEC (not shown).

The 2, 4, and 6 h composite reflectivity forecasts are shown in Fig. 16 for experiments CLIPSAT and 75EnBEC. When the supersaturation removal function is activated in the analyses, a better evolution of the squall line is observed in the 4 and 6 h forecasts (Fig. 16b and c). The displacement errors are reduced and less spurious convection is seen over southern Missouri and northern Arkansas at these forecast hours (Fig. 16b, c, e, and f). As seen in Fig. 15, over these areas the CLIPSAT analysis shows less specific humidity content than in 75EnBEC. However, less spatial coverage of the convection is forecast over eastern

Missouri, and the spurious convection for values lower than 35 dBZ is increased over southwestern Missouri at the 2 h forecast in CLIPSAT when compared to 75EnBEC (Fig. 16a and d). Both experiments overforecast the reflectivity values larger than 35 dBZ over that area, though. The MMI (F+O) values show more skillful forecast of reflectivity larger than 35 dBZ for all forecast lengths in the experiment CLIPSAT. These values are greatly increased at 2 h forecast from 0.540 in 75EnBEC to 0.811 in CLIPSAT due to a better positioning and coverage of the reflectivity above 35 dBZ in areas over southeastern Missouri. Overall, more spurious convection over northwestern Missouri is shown in 75EnBEC which led to the lower MMI (F+O) at this forecast hour (see the blue and red circles over this area in Fig. 16a and d, highlighting improvement and degradation, respectively, for each experiment). At 4 h forecast, MMI (F+O) results show an increase from 0.698 in 75EnBEC to 0.793 in CLIPSAT, with a reduction of the spurious convection between north-central Arkansas and south-central Missouri in CLIPSAT. Nevertheless, most of the spurious convection shown in 75EnBEC over other regions is also observed in CLIPSAT, which may have penalized the MMI (F+O) values in the last experiment. At 6 h forecast, more similar MMI (F+O) values are found, but still less spurious convection is observed for lower reflectivity thresholds in CLIPSAT. Results from CLIPSAT indicate the presence of longer-term bias that is being corrected to some extent in this experiment. However, because the atmospheric state is periodically refreshed with the large scale conditions as part of the partial cycling procedure, the model bias cannot be fully examined. Further investigation involves adapting the approach employed by Wong et al. (2020) in which forecast tendencies are used to investigate systematic model biases in a continuously cycled experiment.

### 4.5 Quantitative Precipitation Forecast Verification

To further evaluate the experiments conducted, the FV3 LAM 1 h accumulated precipitation is also analyzed. Precipitation forecasts remain a challenge for NWP models at various spatial and temporal scales. Because of their complexity, precipitation forecasts are frequently used to evaluate model performance.

As mentioned in Sect. 3.4, precipitation forecasts are verified against Stage IV precipitation observations at various thresholds. Figure 17 shows the ETS and FBIAS for 1 h accumulated precipitation greater than $0.01 \, \text{inches} \, \text{h}^{-1}$ (Fig. 17a and c) and $0.25 \, \text{inches} \, \text{h}^{-1}$ (Fig. 17b and d) for all experiments at each forecast lead hour. These verification measures are based on the two-by-two contingency table used for categorical (dichotomous) variables (e.g., Jensen et al., 2020). ETS is based on the threat score or critical success index and is commonly used to examine the performance of precipitation forecasts. Perfect forecasts have ETS values close to 1, while forecasts without skill have ETS values close to 0. Meanwhile, FBIAS indicates when an event is forecast more or less often than it is observed. FBIAS greater than 1 indicates an event is overforecast, while less than 1 suggests an event is underforecast. An FBIAS equal to 1 indicates that the event is predicted as frequently as it is observed (e.g., Wilks, 2006).

For this case study, ETS values decrease as the precipitation threshold increases in all of the experiments assessed (Fig. 17a and b), indicating the difficulty in predicting heavier precipitation events. Most of the experiments with data assimilation have higher ETS scores for precipitation greater than $0.01 \, \text{inches} \, \text{h}^{-1}$ than NoDA during almost the entire 18 hour forecast (Fig. 17a). This shows the positive impact of data assimilation in the analyses and subsequent lighter precipitation forecasts. Experiments 100EnBEC, CLIPSAT, 75EnBEC, and PSEUDO show higher ETS values in the first 4 hours of the forecast. Be-

tween 4 h and 16 h forecast, experiment CLIPSAT shows the best performance among all experiments, followed by 100EnBEC, 75EnBEC, and PSEUDO, which shows very close results to 75EnBEC. In terms of FBIAS, 100EnBEC shows better scores until the 11 h forecast lead (Fig.17c). Between 2 and 8 h forecast, experiment VLOC shows the greatest underforecast among all experiments. The verification of 1 h accumulated precipitation greater than 0.25 inches consistently shows that using hybrid and pure ensemble BEC in data assimilation improves the precipitation forecasts in the first 13 hours forecast, with 75EnBEC outperforming 100EnBEC within the first four hours (Fig. 17b). After the 13 h forecast, experiment NoDA performs better, which shows data assimilation mainly improves the short term forecast and the major factor for a good long term forecast is the quality of the background from the outside model as well as the FV3 LAM model itself. For the $0.25$ inches h$^{-1}$ threshold, precipitation is overforecast in experiments 100EnBEC and NoDA in the first 3 hours, and underforecast in experiments 3DVar, 75EnBEC, PSEUDO, VLOC, and CLIPSAT. All experiments underforecast accumulated precipitation greater than $0.25$ inches h$^{-1}$ after the 9 h forecast (Fig. 17d).

## 5    Summary and final remarks

The capability of a prototype RRFS with data assimilation, the RRFS v0.1, to simulate convection is investigated through a case study of a squall line that occurred over Oklahoma during the afternoon of 4 May 2020. Various data assimilation parameters and algorithms are tested and evaluated in order to find the best configuration to produce more realistic convection forecasts. This case study shows that the FV3 LAM with the RRFS_PHYv1a physics suite has good potential for storm forecasts. Overall, the configurations tested are able to capture the main characteristics of the major convective systems during the execution period. However, the convection in the RRFS v0.1 tends to be overestimated in intensity and underestimated in its extent, as found in previous studies on FV3-based convection-allowing models (e.g, Tong et al., 2020; Gallo et al., 2021).

As expected, data assimilation makes the analyses fit the observations more closely in all cycles. However, the RMS errors of the OmB show distinguishable spikes in cycles where FV3 LAM 1 h forecasts are initialized from an external model as background for the analyses, which indicates the FV3 LAM is still under spin-up in this situation. Therefore, a cycling configuration including a spin-up period for cycles using external model forecasts may be considered. At present, work is underway at NOAA's Global Systems Laboratory (GSL) and EMC to determine the best cycling strategy for this system.

The data assimilation configurations tested show different impacts to the storm forecasts in terms of the squall line location, orientation and coverage, but experiments with data assimilation show an overall positive impact compared with the experiment without data assimilation. The data assimilation using pure ensemble BEC (100EnBEC) performs better at 2 h forecasts for the storms, but 75 % ensemble BEC (75EnBEC) produces better forecasts in all forecast lengths with a better positioning of the squall line evolution, especially at 4 h forecast. Lower RMSE and bias are also found in experiment 75EnBEC for the analyzed surface variables and most vertical profiles with significant statistically differences below 800 hPa at 00:00 UTC valid hour.

Reducing the vertical localization from 3 layers to 1 layer in the lowest 10 layers of the analysis grid leads to, in general, a less skillful forecast. This suggests that the vertical localization configuration used in RAP is already a good choice and should be used in RRFS. Nevertheless, the RMSE and bias of the 2 h forecast of specific humidity are reduced above 600 hPa at

00:00 UTC valid hour in experiment VLOC as compared with 75EnBEC. In addition, as compared to 75EnBEC, the negative bias present at 1000 hPa in specific humidity and temperature is improved in VLOC as are the high reflectivity values at larger forecast lengths.

Convection is greatly improved when using PBL pseudo-observations from surface 2 m temperature and 2 m moisture observations based on RAP configurations, especially at 4 h forecast with a better coverage and positioning of the convection.
The promising results found in this study for the storm forecast indicates the potential of the PBL pseudo-observations function in future versions of RRFS. The 1 h accumulated precipitation in PSEUDO also depicts this characteristic for most of the forecast hours. The verification of the temperature vertical profiles shows a reduction of bias of the 2 h forecast of temperature below 800 hPa at valid hour 00:00 UTC, but at 12:00 UTC and for specific humidity vertical profiles at both valid hours the experiment PSEUDO shows overall neutral impact over 75EnBEC. On the other hand, wind results show statistically
significant positive impact in the RMSE in the middle atmosphere at valid hours 00:00 UTC and 12:00 UTC. Although, at valid hour 00:00 UTC, the RMSE and bias show degradation below 650 hPa.

Supersaturation clipping in GSI can improve specific humidity fields in the analyses, allowing for more realistic storm and precipitation forecasts at longer forecast lengths. At shorter forecast lead hours, it produces more skillful forecasts with a better positioning and coverage of the reflectivity above 35 dBZ, and precipitation forecasts are as good as in experiments 75EnBEC
and 100EnBEC. Although this function imposes a nonphysical constraint to remove the supersaturation from the background when minimizing the cost function, it leads to overall more skillful forecasts without an increase in the computational cost. These results agree with what is found in previous studies, in which the use of constraints in the analyses led to more skillful forecasts (e.g., Tong et al., 2016, using a divergence constraint). This is a common practice in order to preserve non-negativity in the analyses but also comes at the cost of violating mass conservation (e.g, Janjić et al., 2014, 2021).
FV3 LAM hourly accumulated precipitation forecasts for different thresholds indicate that heavier precipitation ($>0.25$ inches (6.35 mm)) is more difficult to predict than light precipitation ($>0.01$ inches (0.254 mm)). The data assimilation clearly improves precipitation forecasts up to 13 h for both thresholds analyzed. The experiment using 100 % ensemble BEC shows the best 1 h accumulated precipitation forecast quality in the first 4 hours forecast for lighter precipitation, while experiment 75EnBEC performs better for 1 h accumulated precipitation greater than 0.25 inches.
Though this is a single case of a squall line and RRFS components are under development, this study provides valuable insights into the performance of the RRFS v0.1 with various configurations. More extensive testing of RRFS, covering a wider variety of cases, larger domain, and longer period of time, is needed to demonstrate whether results found here are robust or may be case dependent. Although further testing and evaluation are warranted in addition to the options tested here, data assimilation proves to be crucial to improve short term forecasts of storms and precipitation in RRFS.

*Code and data availability.*   The source code repository of the SRW version 1.0.0 is available at https://github.com/ufs-community/ufs-srweather-app (last access: 22 January 2021). The source code of the GSI analysis system used can be found at https://github.com/NOAA-EMC/GSI, branch gsi_fv3reg4coldstart (last access: 26 June 2020). The frozen versions of the codes that comprises RRFS v0.1 can be found at:

https://doi.org/10.5281/zenodo.5546592. As the RRFS is under development, the codes of the different components are constantly evolving. Therefore, for up to date and supported codes, readers are recommended to go to the appropriate GitHub repositories. The MET v9.0.0 source code can be found at https://github.com/dtcenter/MET/tree/main_v9.0. ICs, LBCs, and RAP observations used to perform the experiments and verify the forecasts were obtained from NOAA's High Performance Storage System (HPSS) archives. Stage IV precipitation observations were downloaded from the NCAR Earth Observing Laboratory data server at https://data.eol.ucar.edu/cgi-bin/codiac/fgr_form/id=21.093 (last access: 2 December 2020). Hourly MRMS composite reflectivity mosaic (optimal method) observations are available on the Iowa Environmental Mesonet archives at https://mesonet.agron.iastate.edu/archive/ (last access: 27 February 2021). Storm reports were obtained from the SPC archives available at https://www.spc.noaa.gov/climo/reports/200504_rpts.html (last access: 14 July 2021). The namelist files used for cold or warm start the model, for the analyses in each experiment, and for the generation of the model grid, topography and surface climatology are provided online along with the model configuration file, the file used in the analyses to read the horizontal and vertical scales from an external file, all scripts used to execute every task of the workflow, all scripts used to process model outputs with MET, as well as all scripts and data used to create all figures of the paper, via Zenodo (https://doi.org/10.5281/zenodo.5226389, Banos et al., 2021).

*Author contributions.* IHB set up and performed the simulations, verification and visualization, and prepared the original draft of the manuscript. WM and GG helped to obtain the ICs and LBCs and observational data. GG provided advice on the selection and organization of results and analyses. WM, LFS, and LN provided advice on the verification metrics and the analyses. JRC and LFS helped to conceive the initial idea of this research and provided substantial guidance on the analyses. All authors read, edited, and approved the final manuscript.

*Competing interests.* The authors declare that they have no conflict of interest.

*Acknowledgements.* The authors acknowledge the efforts of many partners at NOAA and NCAR, such as at GFDL, EMC, GSL, and DTC, in developing the RRFS and each of its components, the GSI analysis system, and the MET package. NOAA and NCAR are also thanked for providing access to the computational resources and data used for the development of this research. The authors are grateful to Ming Hu for his guidance and support with RRFS and discussions of the results. The authors also thank Eric Gilleland, Michelle Harrold, and Lindsay Blank for their recommendations on using MET tools and help with statistical discussions. The authors are thankful to Chong-Chi Tong for providing a detailed review of the manuscript before submission and fruitful discussions on the analyses of results. Finally, the authors thank the four anonymous reviewers whose comments and suggestions contributed significantly to the improvement of this manuscript.

Support for this research was provided by the DTC through the DTC Visitor Program. The DTC Visitor Program is funded by NOAA, NCAR, and the National Science Foundation. The first author was financed in part by the Coordenação de Aperfeiçoamento de Pessoal de Nível Superior - Brasil (CAPES) - Finance Code 001. The third author was supported in part by the NOAA Cooperative Agreement with CIRES, NA17OAR4320101.

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

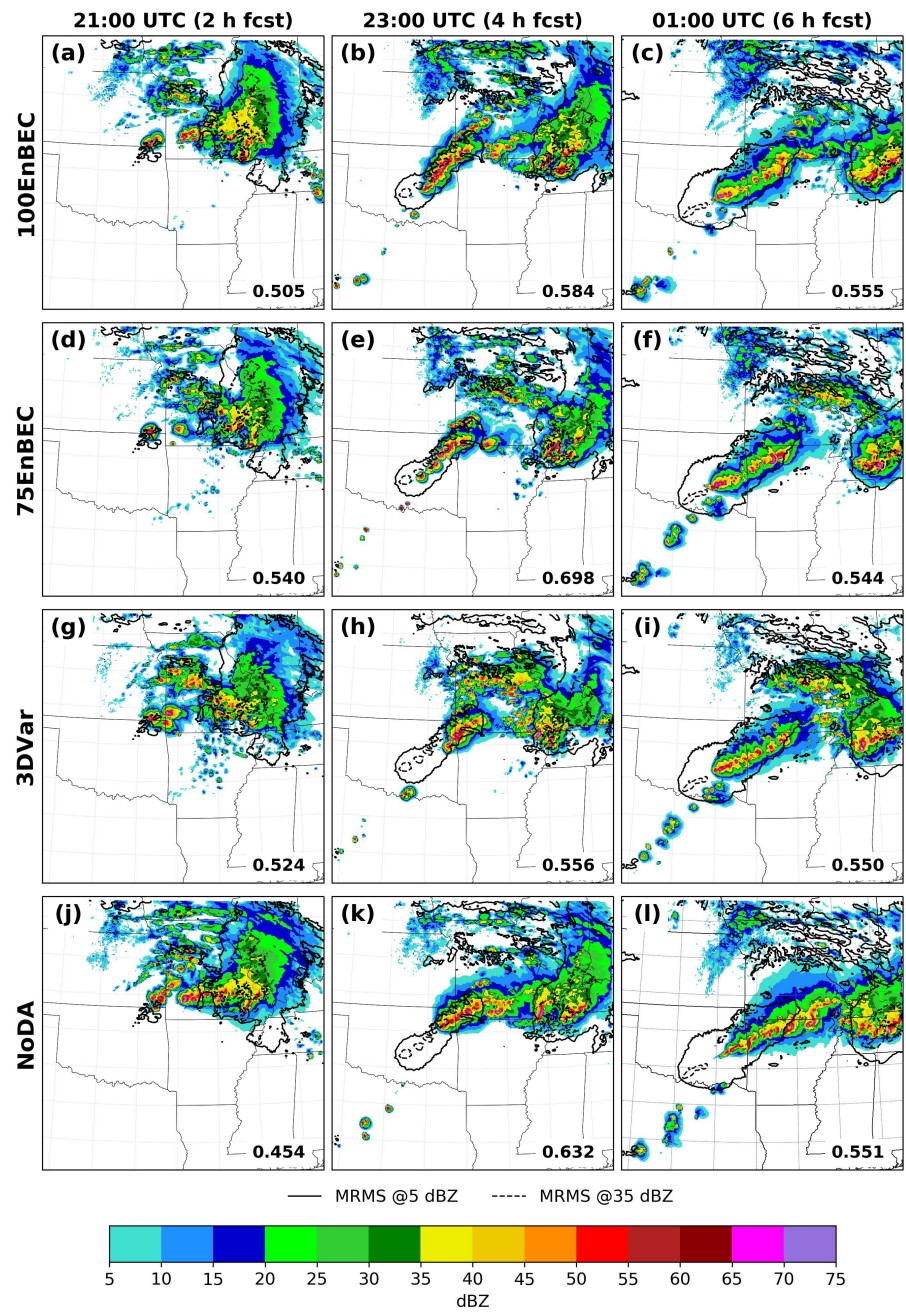

**Figure 7.** 2, 4, and 6 h forecasts of composite reflectivity from experiments 100EnBEC (a, b, and c), 75EnBEC (d, e, and f), 3DVar (g, h, and i), and NoDA (j, k, and l), initialized at 19:00 UTC on 4 May 2020. Solid and dashed black lines are the 5 and 35 dBZ reflectivity observation contours, valid at the forecast time, respectively. MMI (F+O) results for reflectivity values larger than 35 dBZ are shown in the lower right corner of each panel.

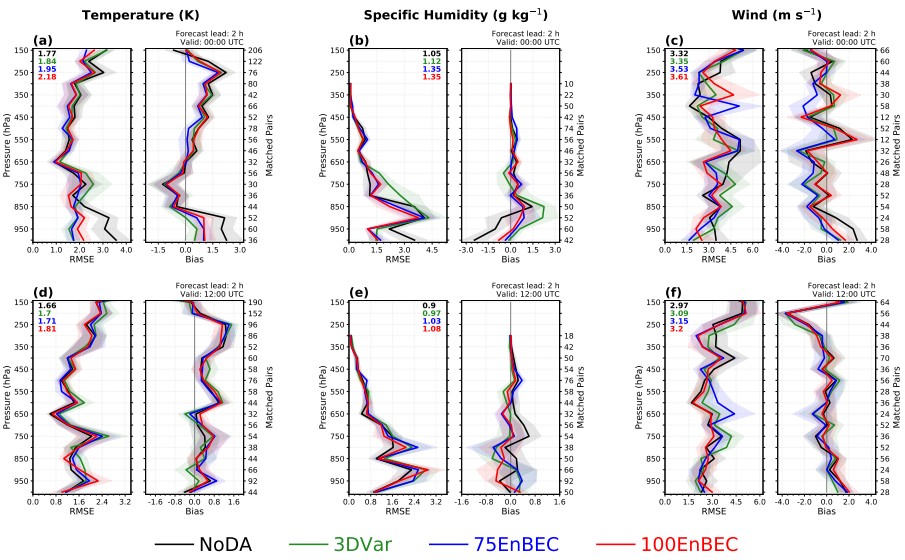

**Figure 8.** Vertical profiles of RMSE (left), bias (right), and upper (95 %) and lower (5 %) limits of the confidence intervals (shading) for the 2 h forecast of temperature (a and d), specific humidity (b and e), and wind (c and f) against rawindsonde, dropsonde, and pilot balloon observations at 00:00 UTC (a, b, and c) and 12:00 UTC (d, e, and f) valid hours on 4 May 2020 for experiments 100EnBEC, 75EnBEC, 3DVar, and NoDA. Matched pair counts used for RMSE and bias computation at each level are shown on the right vertical axis. Each experiment's mean RMSE and bias across all vertical levels are shown in the upper corner of each panel.

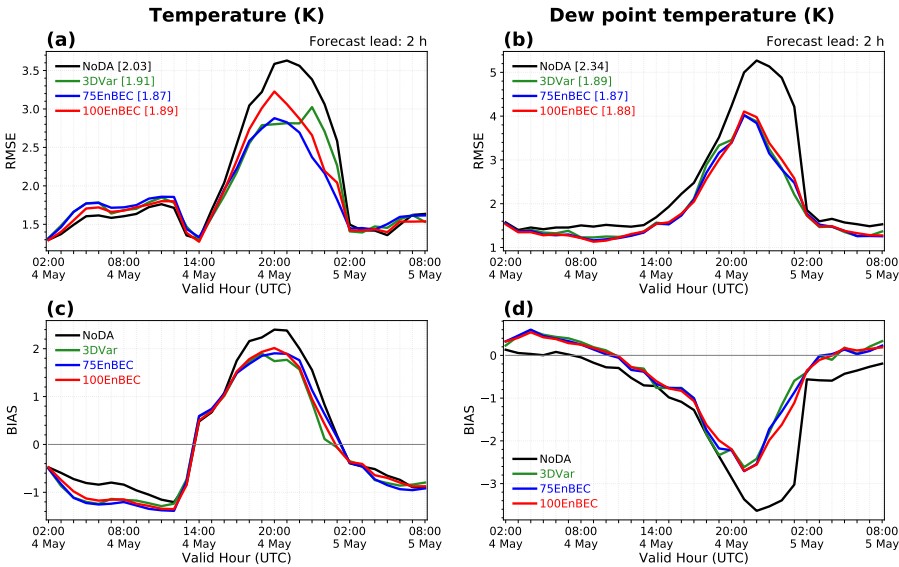

**Figure 9.** RMSE and bias for the 2 h forecast of 2 m temperature (a and c) and 2 m dew point temperature (b and d) against synoptic station and METAR observations for experiments 100EnBEC, 75EnBEC, 3DVar, and NoDA. The legend for each experiment is shown in each panel along with brackets listing the associated RMSE and bias averaged over all cycles.

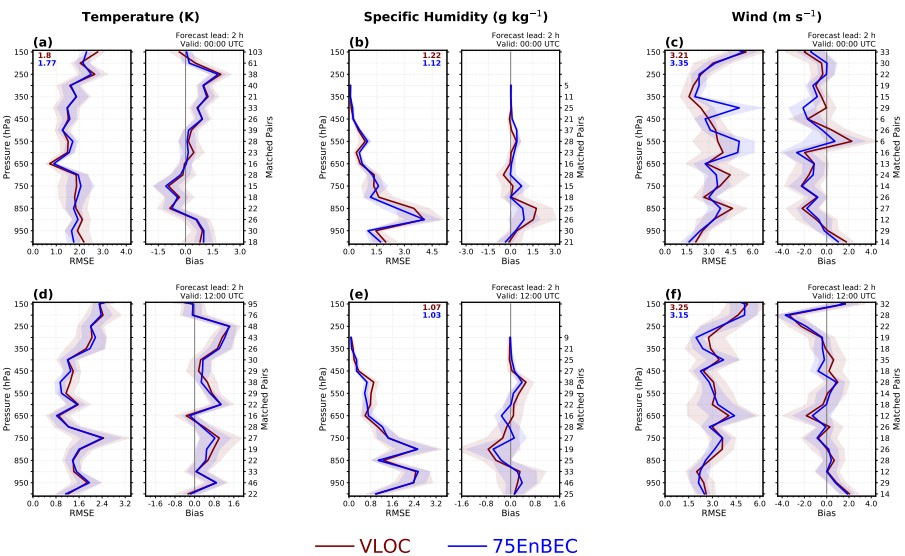

**Figure 10.** As in Fig. 8, but for experiments 75EnBEC and VLOC.

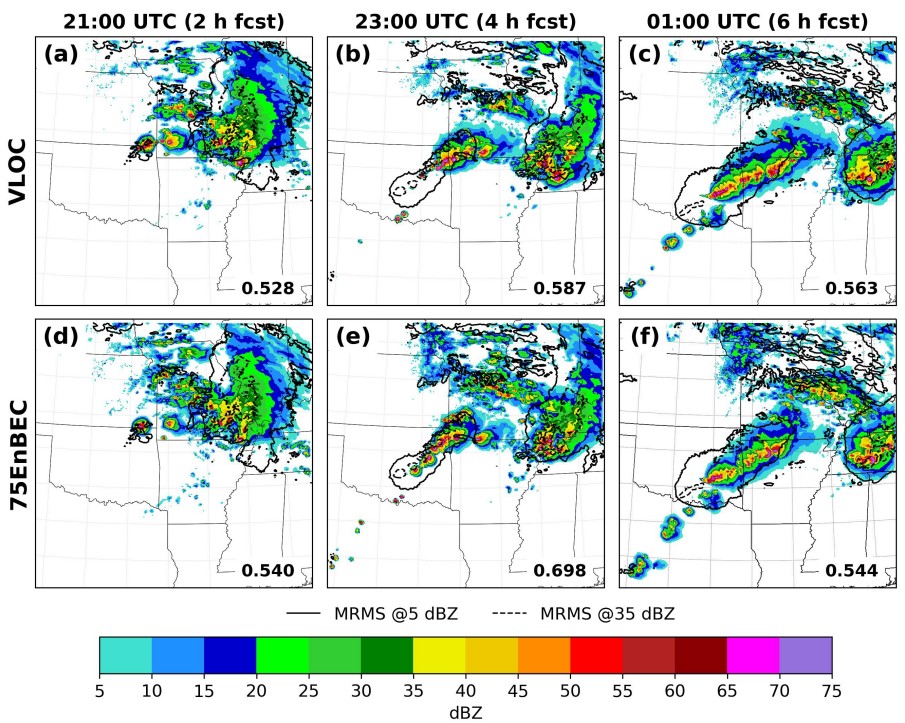

**Figure 11.** As in Fig. 7, but for experiments 75EnBEC and VLOC.

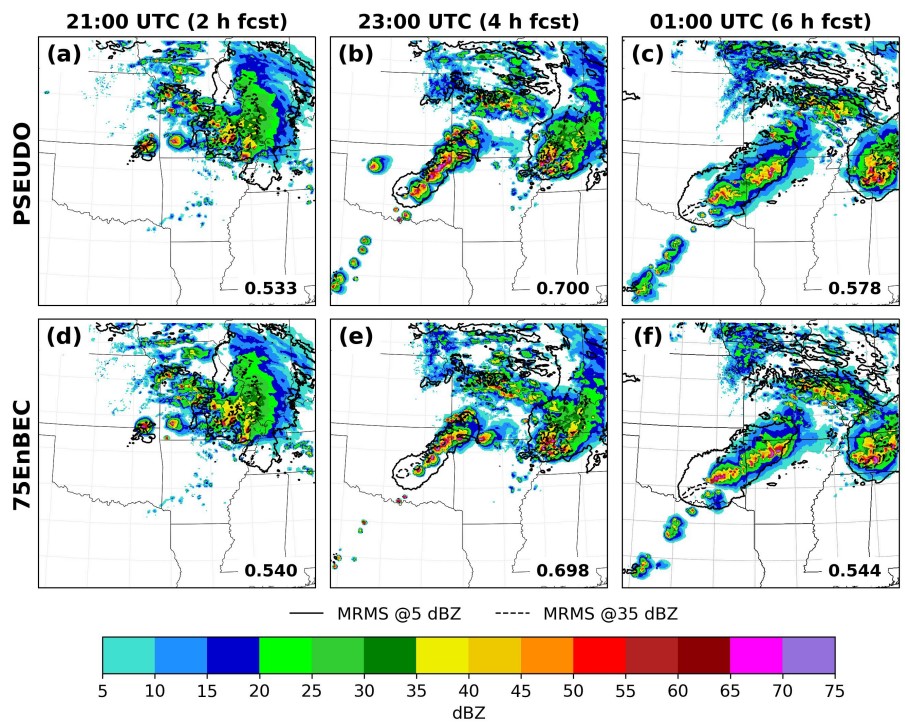

**Figure 12.** As in Fig. 7, but for experiments PSEUDO (a, b, and c) and 75EnBEC (d, e, and f).

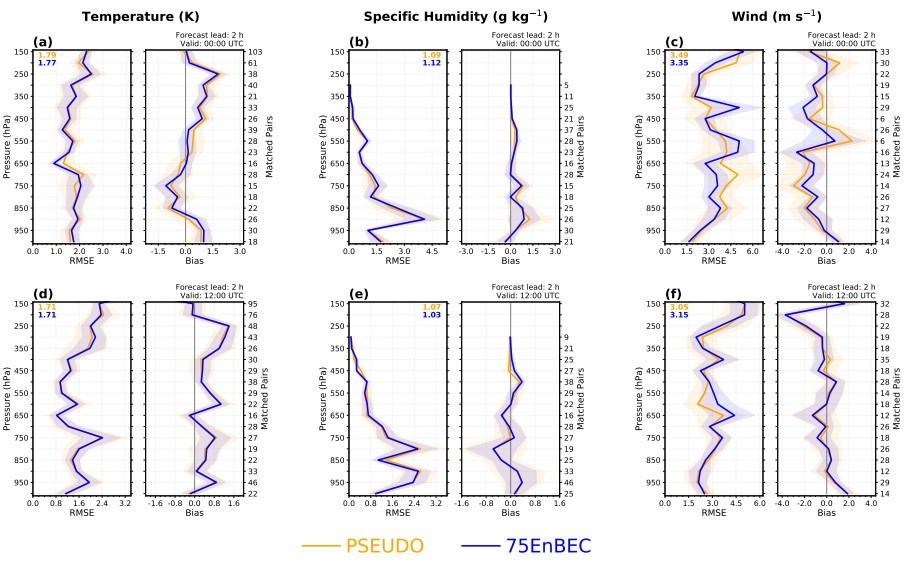

**Figure 13.** As in Fig. 8, but for experiments 75EnBEC and PSEUDO.

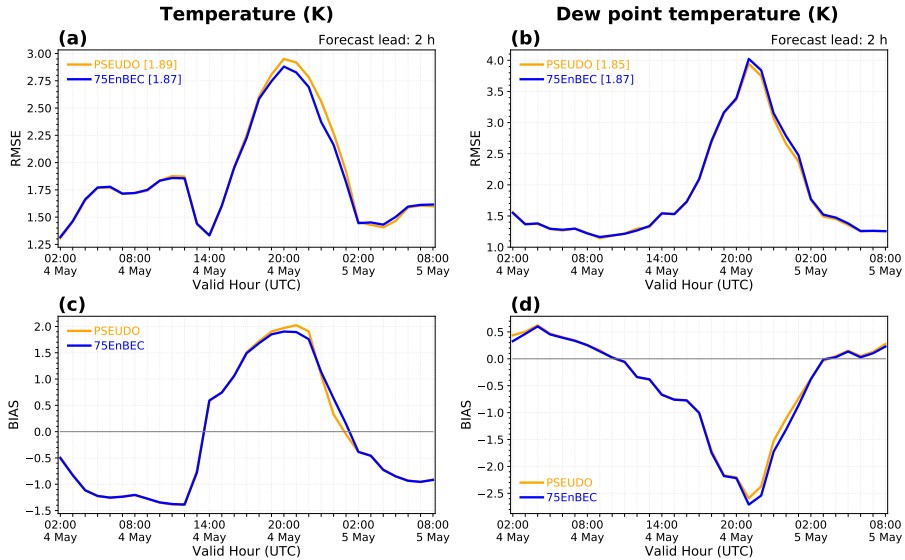

**Figure 14.** As in Fig. 9, but for experiments 75EnBEC and PSEUDO.

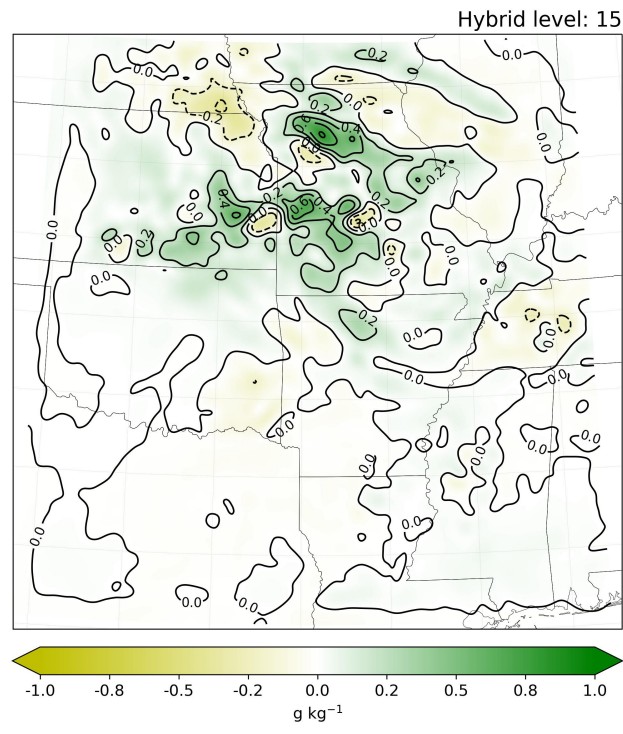

**Figure 15.** Difference in specific humidity ($g\,kg^{-1}$) fields for the 19:00 UTC cycle on 4 May 2020 between analyses without and with supersaturation clipping activated (75EnBEC - CS), at model hybrid level 15.

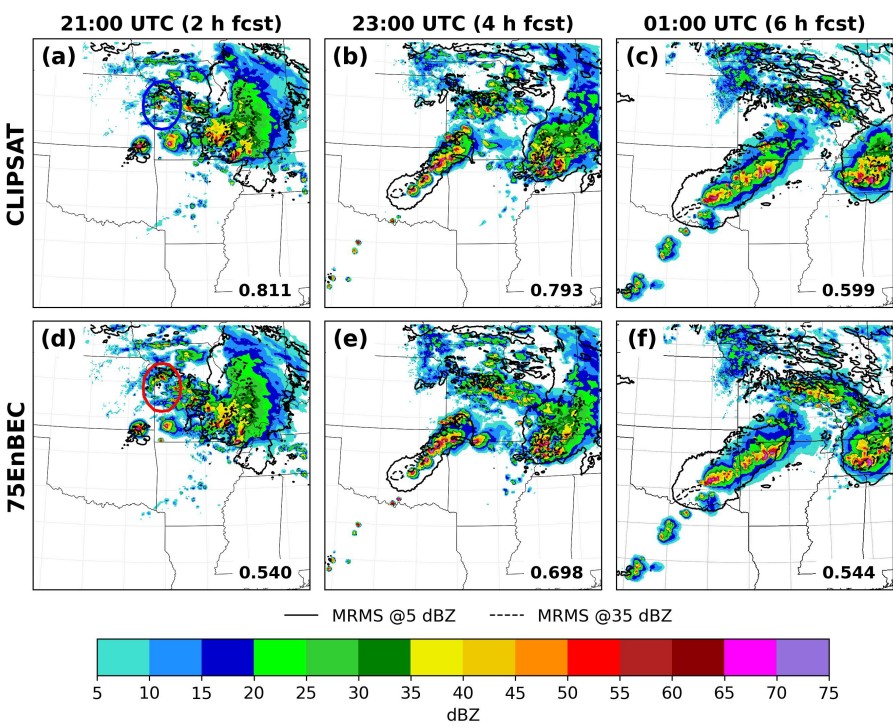

**Figure 16.** As in Fig. 7, but for experiments CLIPSAT (a, b, and c) and 75EnBEC (d, e, and f). The red (blue) circle in panel d (a), respectively, indicates forecast convection that penalized (improved) the MMI (F+O) value in experiment 75EnBEC (CLIPSAT) at 2 h forecast.

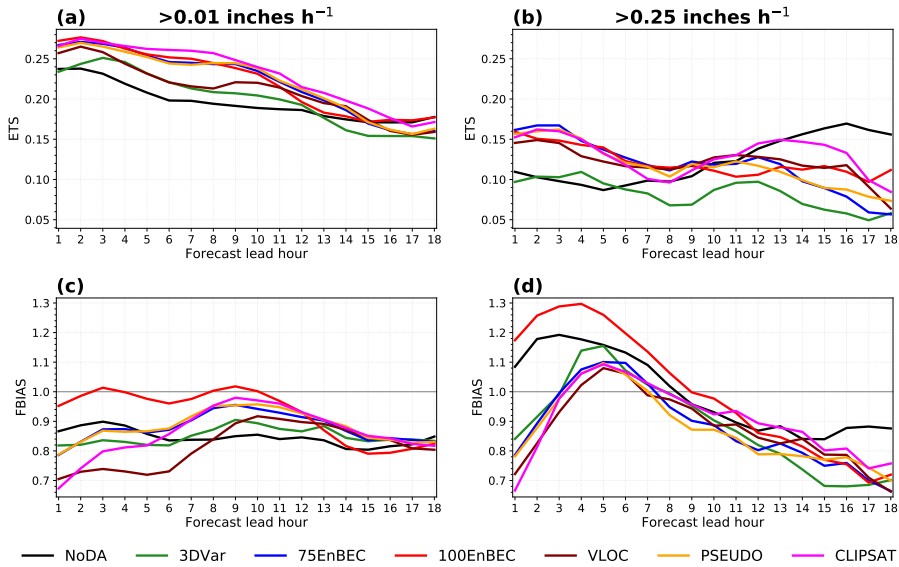

**Figure 17.** ETS (a and b) and FBIAS (c and d) for 1 h accumulated precipitation forecasts greater than 0.01 inches (a and c) and 0.25 inches (b and d) from experiments CLIPSAT, PSEUDO, VLOC, 100EnBEC, 75EnBEC, 3DVar, and NoDA for 18 hour forecasts.