# Peer review of "Assessment of the data assimilation framework for the Rapid Refresh Forecast System v0.1 and impacts on forecasts of a convective storm case study"

_Geoscientific Model Development, 2021_

## Author Response (AR1)

**Reviewer #1**

**General comments:**

**The paper is overall well structured, clearly described, and provides a succinct evaluation of a single use case used for tuning some parameters of RRFS. My one critical comment here is that the title may be a bit misleading, and should be modified if possible (something along the lines of *'... on forecasts of a convective storm case study'*. As it stands, readers are at first likely expecting a larger, more comprehensive, data assimilation evaluation paper consisting of multiple case studies and deeper analyses. To be clear, this single case study paper is useful, but the correct expectation should be set with the title.**

We thank the reviewer for the comments and suggestions. Our responses are noted below. Changes to the document are highlighted in magenta.

The title was modified to "Assessment of the data assimilation framework for the Rapid Refresh Forecast System v0.1 and impacts on forecasts of a convective storm case study"

**Specific comments:**

- **GSI is capable of hybrid 4DEnVar. Is there a reason this flavor of DA was not included in the comparisons?**

  The Rapid Refresh Forecast System (RRFS) is intended for hybrid 3DEnVar data assimilation following the currently operational Rapid Refresh (RAP) configuration. As in RAP, hourly updated cycles are configured in order to leverage available data with higher frequency such as surface observations. Testing and development of a hybrid 4DEnVar system is a worthwhile pursuit, but due to limited resources it is beyond the scope of this article describing the v0.1 capability.

- **For those not familiar with how rapid refresh systems are typically cycled, why is it necessary to perform a periodic cold start even though hourly DA is performed?**

  Rapid refresh systems are usually configured for regional domains in which a reduced amount of data is used compared to global domains. Benjamin et al. (2016) pointed out the importance of a partial cycling technique to update large scale conditions such as the longwave representation. This is of greater importance in regional domains with coverage over oceans such as in RAP and thus in RRFS, where cycles with a warm start from a parallel partial cycle (currently only cold start in RRFS v0.1) are used in order to improve the drift from the global scale. A sentence was added to account for the reviewer's comment.

  Lines 241-243: "Periodic updates of the large scale atmospheric conditions are needed in regional modeling systems in order to account for corrections made by global observations over land and ocean and to avoid model drift from those conditions (Benjamin et al., 2016)."

- **You mention the great importance of tuning localization parameters but only vertical localization is tuned, why is it assumed that the default horizontal localization does not need tuning?**

  We agree with the reviewer. RRFS v1 will use a convective-scale ensemble and therefore the horizontal localization also needs to be tuned in order to assess if the default localization is good enough or if a different value leads to better results. In this study using RRFS v0.1, we use global ensemble members and seek to establish a measure of sensitivity to such parameters to inform further development. For convective systems the vertical representation is very important for storm initialization. Due to limited resources and time, and taking into account that a 3 km grid spacing is used in the experiments, similar to the High Resolution Rapid Refresh (HRRR) horizontal resolution, we decided to only present results of tuning the vertical localization. Testing and evaluating the horizontal localization in RRFS is underway in a separate work. A sentence was added to address the reviewer's comment.

  Lines 312-313: "A separate study is underway in which the optimal horizontal localization for RRFS is also investigated and therefore it is not examined here."

**Technical corrections:**

- **At first I was confused by the different version numbers RRFSv1a/RRFSv0.1, perhaps it would be useful to clarify early on that these are the physics suite / cycling system**

  In order to avoid confusion between these acronyms, RRFSv1a was changed to RRFS_PHYv1a and is more clearly written that RRFS v0.1 is the prototype RRFS used in this study. Lines 67, 111, 152-153, 594 were modified to account for these changes.

  Line 66-67: "...a suite based on GFS version 16 physical parameterizations and a prototype of the RRFS physics suite (henceforth called RRFS_PHYv1a)."

  Line 105: " For the purpose of this paper, the prototype RRFS used is called RRFS v0.1"

- **64 - define what the "convective gray zone" is for readers who might not be unfamiliar**

  Line 65 was modified in order to provide a more straightforward argument of what was investigated in Harrold et al. (2021):

  Line 65: "Harrold et al. (2021) investigated how the SRW represents convection and associated precipitation for varied model grid spacing in two physics suites…"

- **92 - "as good" -> "as well"**

  As suggested, the correction was made in line 94.

Line 94: "…it is imperative that the data assimilation component behave **as well as or better than** the current operational state-of-the-art"

References:

Benjamin, S. G., Weygandt, S. S., Brown, J. M., Hu, M., Alexander, C. R., Smirnova, T. G., Olson, J. B., James, E. P., Dowell, D. C.,Grell, G. A., et al.: A North American hourly assimilation and model forecast cycle: The Rapid Refresh, Monthly Weather Review, 144,1669–1694, https://journals.ametsoc.org/view/journals/mwre/144/4/mwr-d-15-0242.1.xml, 2016

Harrold, M., Hertneky, T., Kalina, E., Newman, K., Ketefian, G., Grell, E. D., Lybarger, N. D., and Nelson, B.: Investigating the Scalability ofConvective and Microphysics Parameterizations in the Unified Forecast System Short-Range Weather (UFS-SRW) Application, in: 101stAmerican Meteorological Society Annual Meeting, AMS, 2021

Reviewer #2
**The paper tested different GSI assimilation settings with FV3 LAM runs, and proved that the system can predict the severe convective squall line case over Oklahoma on 4 May, 2020. The results are useful to most NCEP forecast data users. The paper has some unclear or incomplete reasoning but will likely be a significant contribution with revision and clarification.**

We thank the reviewer for the comments and suggestions. Our responses are noted below. Changes to the document are highlighted in red.

**General comments:**

**For most experiments in this paper, only the comparison results were shown, and the specific causes of the results were not analyzed.**

**For example, the differences between PSEUDO and 75EnBEC experiments were huge, but the authors have not given many diagnostics. Why more observations through GSI will cause overestimated convection?**

- The experiment PSEUDO needed to be rerun because of a bug in the calculation of the planetary boundary layer (PBL) height which affected the PBL pseudo-observations function. This bug only affected this experiment. Accordingly, figures 11, 12, and 13 were replaced with the new results and corresponding discussions and some diagnostics were provided in Sect. 4.3 (lines 512-537) in the new version of the manuscript, as suggested by the reviewer.

**Another example, the VLOC of 1 layer should capture finer vertical features of low atmosphere but the result showed that the positive impact is above 650 hPa and negative impacts are below 800hpa, why?**

- Indeed, the experiment VLOC was conducted in order to capture finer features of the low atmosphere through the reduction of the vertical localization from 3 levels to 1 level in the first 10 model layers. However, the results did not show a positive impact in the lower atmosphere but above 650 hPa. In part this is because the multivariate relationships within the background error covariance that spread the impact to different levels and locations. Also, because of the cycling technique, the impacts in the forecasts are found in other levels. This led us to conclude that the default value of 3 layers already gives the best results in most vertical levels and that the value of 1 layer may be too small considering that the distance between layers in the lower atmosphere is also small. A sentence was added to add more discussion to these results.

  Lines 500-503: "The analysis cycling technique and multivariate relationships in the BEC spread the observations impact throughout different levels and locations, which could have led to the slight positive impact above 650 hPa instead of the lower atmosphere where the modification in the vertical localization was made."

**If the RRFS aims to replace the NCEP operational suite of regional and convective scale modeling systems in the next upgrade, it would be best to show the result from RAP as a baseline for all these tests.**

- This study provides very preliminary results on how the GSI analysis system and the limited area capability of the Finite Volume Cubed Sphere dynamical core (FV3 LAM) performs with the different options tested. It provides an evaluation of different functions and parameter values used currently in HRRR and RAP. However, some functionalities are still being developed and tested and therefore a more comprehensive study with more up-to-date developments of RRFS is underway where a comparison against RAP (as well as HRRR, NAM, and NAM nests) will be shown.

**Specific comments:**

**In Figure 2. RRFS cycling configuration diagram, the cold start is at 0 utc and the warm start seems to be from 1 utc to 6 utc. But in Figure 5 and relative context, the cold start is at 0 utc and 12 utc, what is the exact cold start interval?**

- We agree that the diagram could mislead readers. It was modified to include cycles from 00:00 UTC to 12:00 UTC. Lines 247-250 were modified accordingly.

  Lines 247-250: " Figure 2 illustrates the RRFS cycling configuration from cycles initialized between 00:00 UTC through 12:00 UTC. In each cycle, an 18 h free forecast is launched following the analysis, with hourly outputs. A cold start is performed at 00:00 UTC and 12:00 UTC and warm starts between 01:00 UTC to 11:00 UTC using the FV3 LAM 1 h forecast from the previous cycle as background for the analysis"

**L127 LAM appeared first time, should be limited area modeling (LAM) capability**

- We added the LAM definition right after its first appearance in Line 55, as suggested.

**In Figure 9, no "Matched pair counts used for RMSE and bias computation at each cycle" were found in the photos.**

- We thank the reviewer for pointing out this mistake. The sentence was removed from the caption of Figure 9.

**Reviewer #3**

**General comments:**

**I appreciate the extended introduction and literature review of the state of the modeling and DA development for this application. The manuscript provides a useful snapshot with regards to the state of testing and evaluation of a case study using the RRFS during its development (v0.1).**

**For the benefit of the general data assimilation audience, some detail would be helpful to describe why the unconventional data assimilation cycle is used for the RRFS. For example, why is a cold start applied repeatedly throughout the DA cycles? Why can't the RRFS be run as a continuous cycled process like conventional DA applications? This could perhaps be connected with an improved discussion of the FV3 LAM and its limitations over other possible regional modeling approaches.**

**Please be consistent with the tense throughout. Present tense is appropriate, but occasionally it slips into the use of past tense.**

We thank the reviewer for the comments and suggestions, which were very useful to improve the presentation and scientific quality of the results discussion of this manuscript. Our responses are noted below. Changes to the document are highlighted in blue.

This study represents an evaluation of the initial operating capability of the RRFS. Subsequent development underway includes establishing a partial cycling capability for the inaugural operational implementation, version 1, with subsequent plans to consider a fully cycled version in later versions leveraging recent advances discussed in Schwartz et al. (2021). Lines 244 -247 were modified to include this comment.

Lines 244-247: "Development currently underway includes establishing a partial cycling capability for the inaugural operational implementation, RRFS version 1, with subsequent plans to consider a fully cycled version in later versions leveraging recent advances discussed in Schwartz et al. (2021)."

The verbal tense was corrected throughout the manuscript.

**Specific comments:**

**L 4-5: "The current data assimilation component uses the Gridpoint Statistical Interpolation (GSI) system."**

**It would be helpful to mention here what DA method is being used. Is it 3D-Var?**

- As suggested, the text was modified to include the data assimilation method used, the hybrid three dimensional ensemble–variational data assimilation (3DEnVar):

    Lines 4-5: "The current data assimilation component uses the hybrid three-dimensional ensemble-variational data assimilation (3DEnVar) algorithm in the Gridpoint Statistical Interpolation (GSI) system."

**L 6: "Results show that a baseline RRFS run without data assimilation is able to represent the observed convection, but with stronger cells and large location errors."**

**How does the RRFS represent observed convection without data assimilation? Is it through the boundary/initial conditions coming from the global model? In that case, it would be using data assimilation indirectly through the global analysis. Could the authors please clarify.**

- As we describe in Sect. 3.2 and 3.3, the boundary/initial conditions used in this study are from the 3 km grid High Resolution Rapid Refresh (HRRR) and the experiment without data assimilation uses the same cycling configuration as experiments with data assimilation, in terms of the cold and warm starts. Therefore, it would use data assimilation indirectly through the lateral boundary conditions at each analysis hour and through the cold start initial conditions at 00:00 UTC and 12:00 UTC. Two sentences were added in the abstract and in Sect. 3.3 to clarify this point.

  Line 9: "... without data assimilation..." was removed.

  Lines 7-9: "A domain of 3 km horizontal grid-spacing is configured and hourly update cycles are performed using initial and lateral boundary conditions from the 3 km grid High Resolution Rapid Refresh (HRRR)."

  Lines 299-301: "The 3 km ICs from the HRRR are consistent with the 3 km grid-spacing of the RRFS, such that fine scale features found in the HRRR are present in the RRFS ICs."

**L 8-9: "using 75 % of the ensemble background error covariance (BEC)"**

**What does it mean to use only 75% of the BEC?**

- The information provided was incomplete, the lines were modified to include the weight given to the ensemble and static parts of the background error covariance:

  Lines 10-11:"...especially in the 4 and 6 h forecasts using 75 % of the ensemble background error covariance (BEC) and 25 % of the static BEC…"

**L 9-10: "Decreasing the vertical ensemble localization radius in the first 10 layers of the hybrid analysis results in overall less skillful forecasts."**

**From what initial radius to what final radius?**

**Please change to:**

**"Decreasing the vertical ensemble localization radius [from X m to Y m] in the first 10 layers of the hybrid analysis results in overall less skillful forecasts."**

- The suggested change was adopted:

Lines 12-13: "Decreasing the vertical ensemble localization radius from 3 layers to 1 layer in the first 10 layers of the hybrid analysis results in overall less skillful forecasts."

**L 97-99: "Using hybrid 3DEnVar with 75 % of the ensemble background error covariance (BEC) showed storm structures in the 2 h forecast comparable to when using ensemble Kalman (EnKF), although EnKF outperformed 3DEnVar in the first hour forecast."**

**(1) Does this mean that the BEC is weighted 75% toward the dynamic ensemble-estimated BEC and 25% to the static climatological BEC?**

**(2) If the EnKF outperforms the 3DEnVar in the first hour, and then they are comparable in the second hour, then when not use the EnKF instead of the 3DEnVar?**

- (1) Yes, the text was improved including the weight given to the static BEC:

  Line 100: "... and 25 % of the static BEC…"

- (2) The reviewer would need to follow up with Tong et al. (2020), who conducted the work summarized in this portion of the literature review, to have this question addressed. For RRFS, development is underway to incorporate the EnKF into the hybrid data assimilation system for the first implementation.

**L 99-100: "Both methods showed higher equitable threat scores (ETS) when compared to 3DVar and pure ensemble during the 4 h forecast analyzed."**

**Does "both methods" refer to the 3DEnVar and the EnKF? What is a "pure ensemble"?**

- Yes, "both methods" refer to the 3DEnVar and the EnKF. On the other hand, pure ensemble refers to pure En3DVar in which the BEC is composed of 100 % of the ensemble BEC and 0 % of the static BEC. This was clarified in the text:

  Lines 101-103: "Both methods, hybrid En3DVar and EnKF, showed higher equitable threat scores (ETS) when compared to 3DVar and pure 3DEnVar during the 4 h forecast analyzed."

**L 126-127: "The FV3, originally a global model, features three types of local refinement capabilities: stretching of the global grid (Harris et al., 2016), nesting within the global grid (Harris and Lin, 2013), and a LAM capability (Black et al., 2021)."**

**It would be useful to mention briefly how these types of local refinement differ.**

- The sentence has been updated for additional clarity.

  Lines 129-132: "The FV3, originally a global model, features three types of local refinement capabilities: stretching of the global grid using the Schmidt refinement technique (Harris et al., 2016), one- and two-way nesting within the global grid (Harris and Lin, 2013), and recently a LAM capability (Black et al., 2021). The LAM capability

eliminates the need to run a concurrent global model and instead relies upon lateral boundary conditions provided at pre-specified intervals from an external source."

**L 143-144: "Hence, the CCPP contains a set of physical schemes and a common framework that facilitates the interaction between the physics and a numerical model (Bernardet et al., 2020). "**

**Perhaps it would be more clear to say "between the physics parameterizations and the dynamical core".**

- As suggested, we modified line 148 in the new version of the manuscript.

**L 166: "3D[V]ar"**

- The typo was corrected in line 170.

**L 171: "3DEn[V]ar"**

- The typo was corrected in lines 175 and 180.

**L 177-178: Some discussion should be given about what deficiencies this will have. For example, the lower-resolution global ensemble members (which are even lower resolution than the global deterministic forecast) may have significant biases, and will not resolve the error characteristics at the scale of the LAM. Clearly, the global ensemble statistics provides some useful information, but it is not ideal.**

- As suggested, some discussion was provided in lines 181-184 and lines 185-191.

    Lines 181-184: "Although the use of lower-resolution global ensemble members may not be ideal for the representation of the error characteristics at finer scales, Wu et al. (2017) showed that considerable forecast improvement can be obtained even if the ensemble provided is from a different system, which is consistent with findings in other studies such as Hu et al., (2017)."

    Lines 185-191: "Future work on RRFS involves the extension to a convective-scale ensemble in the EnKF, which will improve the representativeness associated with the forecast error covariance at finer scales. However, such a change is not a panacea. Aside from increased computational expense, the problem of rank deficiency of the ensemble-derived error covariance becomes more apparent with the expanded degrees of freedom associated with the finer spatial resolution. While localization helps somewhat, a computationally affordable ensemble is one that is often insufficiently sized. Therefore future work also includes efforts to introduce multiscale data assimilation capabilities, such as scale dependent localization (e.g., Huang et al., 2021)."

**L 223: "cold starts are performed every 12 hours and warm starts are performed at all other cycles using the 1 h forecast from the previous cycle as background for the analysis."**

**Why is the DA continually reset with cold and warm starts? What prevents the standard self-contained forecast-analysis cycle? Is there a model drift when the DA is run continually in the RRFS?**

- The reviewer is right. Partial cycles are performed in order to avoid model drift when the data assimilation is run continually in the RRFS. Nevertheless, other methods are under consideration for future versions of RRFS, such as a fully cycled method with blending of longwaves from the global based on Schwartz et al. (2022). Lines 241-243 were modified for clarity.

  Lines 241-243: "Periodic updates of the large scale atmospheric conditions are needed in regional modeling systems in order to account for corrections made by global observations over land and ocean and to avoid model drift from those conditions (Benjamin et al., 2016)."

**Figure 2: It would help to have more annotation here. Where are the 18h forecasts coming from?**

- The diagram of Figure 2 was improved showing more cycles in order to clarify that cold starts are performed at 00:00 UTC and 12:00 UTC. The acronym FV3 LAM was added to indicate that the 18 h forecasts are obtained as part of the model execution after the data assimilation.

**L 259:**

**"aircrafts"**

**Change to:**

**"aircraft"**

- The typo was corrected in line 276.

**L 263-264: "The time window used is 1 hour, allowing for observations within 30 minutes before to 30 minutes after the analysis time to be assimilated."**

**I suppose this implies that the analysis time is at the middle of the 1-hour forecast window, but I didn't notice this mentioned earlier.**

- Yes, the time window is centered at the analysis hour. Line 281 includes the word "central" to clarify this information.

**L 265-267: " For the hybrid 3DEnVar analysis, the Global Data Assimilation System (GDAS) 80 member ensemble forecasts (9 h forecasts) are used to provide the ensemble BEC (e.g. Wu et al., 2017)."**

**Please mention the resolution of these ensemble members. How well do the lower resolution members resolve dynamics at the scale within the 3km resolution LAM? E.g. how many grid points of the low-res FV3 global ensemble member fall within the LAM region?**

- The resolution of the 80 member ensemble forecast is approximately 25 km. As mentioned in a previous comment, the use of lower-resolution global ensemble members may not be ideal for the representation of the fine scale motion seen at 3 km. However, doing so has shown to be beneficial (Hu et al. 2017) for the forecasts. It should be taken into account that this study uses a prototype RRFS, RRFS v0.1 and that future work already intends to address this issue. This was added in lines 181-184 and 288-289. Line 284 was also modified to include the resolution of the ensemble members, as suggested.

  Lines 288-289: "As shown in Hu et al. (2017), using off-time global and fixed ensemble based BEC still produces better results than just using the static BEC."

  Line 284: "These forecasts have a horizontal resolution of approximately 25 km and…"

**L 268-269: "For example, the 9 h GDAS ensemble forecasts initialized at 00:00 UTC (valid at 09:00 UTC) are used for the cycles from 07:00 UTC to 12:00 UTC."**

**How is the 9-hour forecast initialized at 0 UTC used at 12 UTC? Is this using the BEC fixed in time at 9 UTC for the entire window?**

- Yes, since GDAS ensemble forecasts are available four times per day, the same ensemble based BEC is used along the 6 h window. This is possible because GSI has the flexibility to use off-time ensemble forecasts, i.e. ensemble forecasts that do not match the analysis hour. As shown in Hu et al. (2017), using off-time global and fixed ensemble forecasts still produces better results than just using the static BEC. The following sentence was added to the manuscript:

  Lines 288-289: "As shown in Hu et al. (2017), using off-time global and fixed ensemble based BEC still produces better results than just using the static BEC."

**L 271-272: "In all experiments with data assimilation, two outer loops with 50 inner loops each are performed to minimize the cost function and find each analysis."**

**I'm not sure I understand what is done in the outer and inner loops. Are the inner loops referring to the PCG solver? If so, then was is done in the outer loop?**

- In this study, the minimization was performed in two outer loops with 50 iterations each, which were sufficient to reach the convergence condition. In each outer loop a re-linearization is performed after the 50 iterations. The increment is zero for the first outer loop while for the second it is updated with the solution found after the 50 iterations of the first outer loop.

  Lines 289-292: "In all experiments with data assimilation, two outer loops with 50 iterations each loop are performed to minimize the cost function and find each analysis. In each outer loop a re-linearization is performed (e.g., Kleist et al., 2009). The increment is zero for the first outer loop while for the second it is updated with the solution found after the 50 iterations of the first outer loop."

**L 274-275: "for different [applications]."**

- As suggested, "practices" was changed to "applications" in line 296 of the new manuscript.

**L 277-279: "This baseline experiment is called NoDA and uses the same cycling configuration as experiments with data assimilation but without the execution of GSI."**

**I understand that from your perspective this doesn't use the GSI, but for the entire procedure the GSI is used at the global scale. I think it would be helpful to explain this context in more detail.**

- The sentence was modified in line 299 to clarify that the cycling configuration of the NoDA experiment follows the same configuration of the experiment with data assimilation, in terms of the cold and warm initial conditions. Explanation of what is performed in each initialization type is already provided in Sect. 2.7.

**L 281: "experiments with different ensemble weights [are] conducted"**

- The verbal tense was corrected in line 303 of the new manuscript.

**L 335: "The analysis residuals (OmA) are also depicted in Fig. 4"**

**The OmA's are less useful than the OmF's for assessing the DA performance. The analysis can be drawn arbitrarily close to the observations (e.g. using complete replacement). It is more valuable to see that the forecasts are being drawn closer to the future observations.**

- The reviewer is right. However, for a more complete assessment, results from both, OmA and OmF, are provided in Figure 5 for all cycles. Figure 4 intends to show the spatial distribution of the assimilated observation and inform readers on the performance of the data assimilation system.

**L 343: "while some MESONET observations have large analysis residuals."**

**What is the expected cause of this larger discrepancy in these observations?**

- As pointed out in Morris et al., (2020), while some MESONET stations are well maintained, the majority do not meet siting standards and maintenance protocols. Therefore, larger residuals are expected from these observations when compared to other observation networks such as METAR, which is an officially maintained observation network. Owing to their uncertain quality, MESONET observations are therefore assigned a higher observation error via a station blacklist, which also explains the larger residuals. Lines 373-376 were modified to include more details.

  Lines 373-376: "As pointed out in Morris et al., (2020), while some MESONET stations are well maintained, the majority do not meet siting standards and maintenance protocols and therefore are assigned a higher observation error via a station blacklist. As expected, larger residuals are found from these observations when compared to other observation networks."

**L 350: "This means the analyses fit closer to the observations, which is expected from a correctly executed data assimilation procedure."**

**This is partially true, but it is also not correct to fit the observations perfectly (due to observational error), so care should be taken in such a statement.**

- We agree with the reviewer. The sentence was modified to:

  Lines 383-384: "This means the analyses fit the observations more closely, though owing to observation error not perfectly, which is expected from a correctly executed data assimilation procedure."

**L 351-353: "There is a noticeable jump in the RMS error values of the OmB from 00:00 UTC (12:00 UTC) to 01:00 UTC (13:00 UTC) on 4 May 2020. This is because 00:00 UTC and 12:00 UTC are cold started from HRRR analyses. "**

**So why are you using the cold starts?**

- As explained before, the cold starts in this study are basically to update the large-scale conditions and to correct the drift from the model. A more adequate cycling technique is being investigated as stated in lines 601-602 in the manuscript.

**Figure 5: It would be useful to add a panel showing the difference between 3dvar and the 75EnVar.**

- In the discussion provided for Figure 5 it is highlighted that the background root mean square error increases from 14:00 UTC to 23:00 UTC, and that in Figure 5b this increase is larger than in Figure 5a (3DVar vs. 75EnBEC). Therefore, arrows were added to Figure 5 to draw attention to that marked increase, which is the focus of this result. The differences for other periods of time are subtle between these two panels. Accordingly, the caption of Figure 5 was updated.

  Caption: "Black arrows highlight the time period from 14:00 UTC to 23:00 UTC."

**L 365: "the EnKF filter method."**

**Change to:**

**"the EnKF method."**

- The word "filter" was removed from line 398 in the new version of the manuscript.

**L 377: "Figure 7 shows the 2, 4, and 6 h forecasts of …"**

**The analysis of these results seems very subjective - could the authors please provide some general statistics for each experiment result to help in the comparison, and provide more objectivity. Below, for example, I suggest adding RMSE statistics to each sub-plot in Figure 7.**

- The reviewer is right. In the new version of the manuscript, the Method for Object-Based Diagnostic Evaluation (MODE) in the Model Evaluation Tools (MET)

(Jensen et al., 2020) was used for reflectivity values larger than 35 dBZ. The median of maximum interest (MMI (F+O)) (Davis et al., 2009) derived from MODE results were then added to each subplot. This metric is an appropriate metric for this kind of verification, providing more objectivity to the discussion, as suggested by the reviewer. Lines 352-359 were added to the document in which the method and metric used are described. Corresponding discussion of these values was added to the new version of the manuscript in figures 7, 11, 15, and A1.

Lines 353-359: "Hourly MRMS composite reflectivity mosaics (optimal method) observations (Zhang et al., 2016) are used to verify the composite reflectivity forecasts using the Method for Object-Based Diagnostic Evaluation (MODE) in MET. In order to quantitatively identify the experiment configuration that yielded better forecasts, the median of maximum interest (MMI (F+O)) (Davis et al., 2009) is analyzed. This metric results from the median between the maximum interest from each observed object with all predicted objects (MIF), and the maximum interest from each predicted object with all observed objects (MIO). It takes into account all attributes used in the total interest calculation, summarizing them into a single value. The forecast in greatest agreement with the observations will give MMI (F+O) values closer to one. Otherwise, the values will be closer to zero."

**Figure 6: Where does the error covariance matrix come from for the 3D-Var? How was it computed?**

- The static BEC for the 3DVar is the same currently used in RAP and HRRR. It is based on statistics from the Global Forecast System (GFS) and the North American Mesoscale Forecast System (NAM) forecasts. Line 306 was added to the new version of the manuscript.

  Line 306: "The static BEC for the 3DVar is the same as currently used in RAP and HRRR (Benjamin et al. 2016)."

**L 405-407: "The wind RMSE results do not clearly indicate which experiment is best, but in general 100EnBEC shows the lowest values when considering all vertical levels."**

**The hybrid methods generally underperform when the static BEC is inadequate. How is the BEC computed for these experiments? This may indicate that tuning is necessary for the B matrix in the hybrid, e.g. see:**

**Chang et al., 2020:**

https://journals.ametsoc.org/view/journals/mwre/148/6/mwrD190128.xml

- We agree with the reviewer, the static BEC matrix used may not be optimal for RRFS and efforts are underway in order to obtain this matrix using its own RRFS forecasts. As mentioned above, the static B used in the experiments is the one currently used in RAP and HRRR.

Line 450-451: "These results may indicate that the static BEC matrix used may not be optimal for RRFS v0.1 and efforts are underway in order to obtain a better BEC matrix."

**Also, the online estimation of the hybrid weighting parameter has been explored by De Azevedo et al. 2020, which may be worth mentioning:**

https://www.tandfonline.com/doi/full/10.1080/16000870.2020.1835310

- As suggested, the work of Azevedo et al. (2020) was mentioned in lines 451-453.

  Lines 451-453: "Moreover, an online estimation approach may be explored for the specification of the hybrid weighting parameter, such as the method proposed by Azevedo et al. (2020) in which a geographically varying weighting factor alpha is defined and the ensemble spread is used for the assignment of the weights."

**Figure 7: It would be helpful if each one of these plots had an RMSE value appended (e.g. in the lower left corner) to make it easier to compare the methods.**

- As mentioned previously, in the new version of the manuscript, the Method for Object-Based Diagnostic Evaluation (MODE) in the Model Evaluation Tools (MET) (Jensen et al., 2020) was used for reflectivity values larger than 35 dBZ. The median of maximum interest (MMI (F+O)) (Davis et al., 2009) derived from MODE results were then added to each subplot. This metric provides more objectivity to the discussion, as suggested by the reviewer. Lines 353-359 were added to the document in which the method and metric used are described. Corresponding discussion of these values was added to the new version of the manuscript in figures 7, 11, 15, and A1.

**Figure 8: The Green RMSE is difficult to read. A slightly darker color would help. Also, perhaps you could order the RMSE in each plot from lowest to highest so that is it easier to see how each method performs in comparison to the others.**

- As suggested, Figure 8 was improved using a darker green for the 3DVar experiment results and the RMSE values were ordered from lowest to highest. Figures 9 and 16 were also modified to account for the color change in the 3DVar plot line.

**L 430: "this study looked at"**

**Change to:**

**"this study looks at"**

- The verbal tense was corrected in line 476 of the new version of the manuscript.

**Figure 8 and 10: I'm not sure that I see the value of the confidence interval shading. However are these confidence intervals computed? Can the authors justify that this statistic is meaningful for this application? (e.g., the confidence interval implies that this specific case can be extrapolated to all other relevant storm instances - perhaps**

**more discussion of the computation and application of this method would be warranted.)**

- We agree with the reviewer that the confidence intervals interpretation was not added to the discussion and therefore it was improved in the new version of the manuscript. The confidence intervals used in this study were derived using a bootstrap resampling technique of 1000 replications with replacement at each forecast lead hour in every cycle, and with bias-corrected percentiles, as reported by Wilks, 2006 and Gilleland et al. (2018). The confidence intervals help to highlight where the differences between the experiments analyzed are significant or not. Adding the confidence intervals shading shows the variability in the sample and informs readers how confident we can be that the mean values plotted are actually different. However, since we are analyzing multiple forecasts for this one single case study, we are not confident that the results obtained in this study can be extrapolated to all other relevant storm instances. Nevertheless, some of the results show statistically significant differences for the sample of this case study, which were highlighted in the discussions of figures 8, 10, and 12 in the new version of the manuscript. As suggested, several lines in the document (435-444, 487-491, 526-529) were modified for clarity and to address the point made by the reviewer.

**L 450-451: "The impact of adding PBL pseudo-observations to the analysis based on surface temperature and moisture observations is evaluated in experiment PSEUDO."**

**I may have missed it, but I don't see the source of the pseudo-observations. What measurements are being converted to these pseudo-ops? Is there a reason the source measurements cannot be assimilating using an appropriate observation operator?**

- This function is currently used in operational RAP and HRRR. In Sect. 3.3, we provided readers a background on why this function was developed and referenced the studies of Benjamin et al., 2004, Smith et al. (2007), Benjamin et al. (2010), and Benjamin et al. (2016) where more details on this function can be found. For clarity, line 507 was updated.

  Line 507: "...using METAR observations,…"

**L 477: "More tuning and testing of this function are needed before applying this technique in the RRFS."**

**The results (e.g. in Figure 12) seem to indicate the use of pseudo-observations may be a bad idea. Is there any reason why continued effort should be made to do this?**.

- The experiment PSEUDO needed to be rerun because of a bug in the calculation of the planetary boundary layer (PBL) height which affected the PBL pseudo-observations function. This bug only affected this experiment. Accordingly, figures 11, 12, and 13 were replaced with the new results, which indicate that this strategy improves the convection in particular for the 4 h forecast. The new analysis of these results  corresponding discussions were provided in Sect. 4.3 (lines 512-537) in the new version of the manuscript.

**Figure 13: Based on Figure 12, this seems to be over-fitting the data, which is then causing problems throughout the model column. It seems like this result would be better achieved via post processing, so that it doesn't have negative ripple effects throughout the cycled DA.**

- Figure 13 was updated with the new results and corresponding discussion was provided.

**Figure 14: Please state what the units and physical height correspond to model level 50.**

- The model level 50 is a hybrid level which corresponds with around 850 hPa at approximately 1450 m of altitude. As suggested, line 543 was modified for clarity as well as "Hybrid" was included in Figure 14.

  Line 543: "...model hybrid level 50 (located in the lower atmosphere at around 850 hPa)"

**L 559: "Supersaturation clipping in GSI can improve specific humidity fields in the analyses, allowing for more realistic storm and precipitation forecasts at longer forecast lengths. At shorter forecast lead hours, it produces more spurious convection"**

**This sounds like the wrong modification is being made to correct a longer-term bias. It would be beneficial to track down the root cause that accumulates to cause the longer-term bias without degrading the short-term skill.**

- We agree, however this also highlights a challenge with the partial cycling procedure where we cannot fully examine the model bias owing to the fact that the atmospheric state is periodically refreshed with the global models. Future work involves adapting the approach employed by Wong et al. (2020).

  Lines 556-560: "Results from CLIPSAT indicate the presence of longer-term bias that is being corrected to some extent in this experiment. However, because the atmospheric state is periodically refreshed with the large scale conditions as part of the partial cycling procedure, the model bias cannot be fully examined. Further investigation involves adapting the approach employed by Wong et al. (2020) in which forecast tendencies are used to investigate systematic model biases in a continuously cycled experiment."

**L 572: "More extensive testing of RRFS, covering a wider variety of cases, larger domain, and longer period of time, is needed to demonstrate whether results found here are robust or may be case dependent."**

**I agree, which is why I'm confused by the presentation of the confidence intervals. Could the authors either remove these in the plots above or describe in greater detail how they were computed and why they are relevant in this case.**

- As mentioned above, we agree with the reviewer that the confidence intervals interpretation was not included in the discussion and therefore it was improved in the

new version of the manuscript. It is important to highlight that the results presented are representative for the case study analyzed. However, we are not confident that the results obtained in this study can be extrapolated to all other relevant storm instances. Therefore we conclude that more tests covering a wider variety of cases is needed in order to obtain more robust results.

References added to the manuscript in function of this comments and suggestions:

Azevedo, H. B. D., Gonçalves, L. G. G. D., Kalnay, E., and Wespetal, M.: Dynamically weighted hybrid gain data assimilation: perfect model testing, Tellus A: Dynamic Meteorology and Oceanography, 72, 1–11, https://doi.org/10.1080/16000870.2020.1835310, 2020.

Davis, C. A., Brown, B. G., Bullock, R., and Halley-Gotway, J.: The Method for Object-Based Diagnostic Evaluation (MODE)Applied to Numerical Forecasts from the 2005 NSSL/SPC Spring Program, Weather and Forecasting, 24, 1252–1267,https://doi.org/10.1175/2009WAF2222241.1, 2009.

Derber, J. and Rosati, A.: A global oceanic data assimilation system, Journal of physical oceanography, 19, 1333–1347, 1989.

Schwartz, C. S., Poterjoy, J., Carley, J. R., Dowell, D. C., Romine, G. S., and Ide, K.: Comparing Partial and Continuously Cycling Ensemble Kalman Filter Data Assimilation Systems for Convection-Allowing Ensemble Forecast Initialization, Weather and Forecasting, 37, 85 – 112, https://doi.org/10.1175/WAF-D-21-0069.1, 2022

Morris, M. T., Carley, J. R., Colón, E., Gibbs, A., Pondeca, M. S. F. V. D., and Levine, S.: A Quality Assessment of the Real-Time MesoscaleAnalysis (RTMA) for Aviation, Weather and Forecasting, 35, 977 – 996, https://doi.org/10.1175/WAF-D-19-0201.1, 2020.

Wong, M., G. Romine, and C. Snyder, 2020: Model improvement via systematic investigation of physics tendencies. Monthly Weather Review, 148, 671-688. https://doi.org/10.1175/MWR-D-19-0255.1

Zhang, J., Howard, K., Langston, C., Kaney, B., Qi, Y., Tang, L., Grams, H., Wang, Y., Cocks, S., Martinaitis, S., Arthur, A., Cooper, K.,Brogden, J., and Kitzmiller, D.: Multi-Radar Multi-Sensor (MRMS) Quantitative Precipitation Estimation: Initial Operating Capabilities,Bulletin of the American Meteorological Society, 97(4), 621–638, https://doi.org/10.1175/BAMS-D-14-00174.1, 2016.

---

## Referee Report (RR1)

**Review of "***Assessment of the data assimilation framework for the Rapid Refresh Forecast System v0.1 and impacts on forecasts of a convective storm case study***"**

The authors use a relatively new modeling system to produce a set of forecasts of a convective-scale event. Sensitivity experiments are performed to assess the impact of different data assimilation choices with standard verification metrics. The authors do a nice job of explaining the new system, and providing justification for their choices, as well as incorporating a diversity of verification approaches. My major concerns with the study are the overlap with prior work (e.g., Tong et al. (2020) used similar model and DA systems) and the lack of additional cases for analysis. I've provided some minor comments below that I believe need to be addressed before recommending acceptance.

**Minor comments**

- I suggest the authors provide more detail about how this study differs compared to Tong et al. (2020), especially the model configuration and design choices. There are a lot of similarities, including the use of FV3 (termed the FV3-SAR in that study, which I believe is the same model that is the core component of the UFS-SRW), variational and hybrid DA with GSI, similar physics choices, and a similar convective-storm case study approach (although for a different case). Can the authors describe how that work ties in with the current set of experiments?

- The RRFS will be an ensemble-based system, so generating and verifying ensembles seems like a good choice to assess the benefits of the various approaches. It may be useful to clarify why the authors only performed deterministic forecasts somewhere in the text (sorry if it's there and I missed it!).

- I really think this study would benefit from additional cases, especially when the authors argue at many points in the paper they are using the results to guide future configuration decisions. Some of the differences between the experiments seem very small, and may become more evident with a larger sample size. This could be considered a "fatal flaw" by some, but I think there's some merit in providing documentation of ongoing work leading up to the implementation of the future RRFS system in the form of this manuscript.

**Specific comments**

Lines 40-42: This sentence implies that the addition of the WaveWatch model into the operational forecast somehow improved the low-level cold temperature bias observed in a prior version of the GFS. I don't think that's possible and I don't think the change notice referenced supports that claim. Please revise.

Line 87-91: I recommend removing these sentences. The number of studies that examine data assimilation for convection-allowing applications is too numerous to mention here, so describing

these two specific studies is necessary, unless they are aspects of the work that are especially relevant to the current work.

Line 104: The authors should make clear that the eventual RRFS implementation will produce ensemble forecasts and not just a single deterministic forecast.

Line 153-161: I suggest moving the list of these parameterizations into a Table that can be referenced in the future, including names of schemes and associated studies that describe each scheme.

Section 2.6: What do the authors mean by "workflow"? As written, the term is used rather generically, but I'm guessing that there is specific workflow software that is used that should be described in the text (this may be described later in the text, but the authors should bring this up earlier).

Line 260-262: How were the MLCAPE and shear diagnostics computed? The text states they were "observed", but there are no routine soundings typically available between 19-20 UTC in northeastern Oklahoma. If these values are from a model analysis, that should be stated (e.g., "The RAP analysis contained MLCAPE values of…).

Line 367: Are Oklahoma Mesonet observations assimilated?

Line 543: Is model level 50 really located around 850 mb? Does that mean that there are 50 levels below 850 mb and only 14 levels above 850 mb (64 levels total)?

Lines 554-556: How are the MMI values so different at 21 UTC in CLIPSAT and 75EnBEC? To my eye, the figures look almost identical. The differences at 23 UTC look more significant, but the MMI values are more similar at this time. Can the authors explain why this is the case?

Figure A1: Why is this included as figure A1 and not Figure 11?

---

## Author Response (AR2)

**Review of "*Assessment of the data assimilation framework for the Rapid Refresh Forecast System v0.1 and impacts on forecasts of a convective storm case study*"**

**The authors use a relatively new modeling system to produce a set of forecasts of a convective-scale event. Sensitivity experiments are performed to assess the impact of different data assimilation choices with standard verification metrics. The authors do a nice job of explaining the new system, and providing justification for their choices, as well as incorporating a diversity of verification approaches. My major concerns with the study are the overlap with prior work (e.g., Tong et al. (2020) used similar model and DA systems) and the lack of additional cases for analysis. I've provided some minor comments below that I believe need to be addressed before recommending acceptance.**

We thank the reviewer for the comments and suggestions, contributions which improve the quality of the final version of the manuscript. Our responses are noted below. Changes to the document are highlighted in blue.

Indeed this study has some overlap with prior work such as Tong et al. (2020). As pointed out by the reviewer, the main overlapping is regarding the use of the limited area model (LAM) capability based on the Finite-Volume Cubed-Sphere (FV3) dynamical core (FV3LAM) and the Gridpoint Statistical Interpolation (GSI) analysis system. However, it should be noted that, up to the moment this manuscript was submitted, the only study in which both of these systems were used is Tong et al. (2020). Therefore, the present study intends to fill part of the gap in the literature by providing a description of the initial data assimilation infrastructure and assessing the analyses and forecasts produced with the prototype RRFS system. Some of the functionalities and configurations tested, such as the planetary boundary layer (PBL) pseudo-observations function of GSI, the localization scale radius, and cycling strategy, are based on currently operational Rapid Refresh (RAP) and High Resolution Rapid Refresh (HRRR) modeling systems. Thus, this study informs developers of a set of configurations that lead to better convection forecasts and points out aspects that require more tuning and improvements inside the prototype system. We are aware that additional case studies covering a wider variety of convection modes and time frames are needed in order to obtain more robust results, which is the subject of future studies. However, and as mentioned in the document, we intended to explore more functionalities and configurations on a single case in order to establish an understanding of baseline sensitivities, which can help future RRFS implementations.

**Minor comments**
- **I suggest the authors provide more detail about how this study differs compared to Tong et al. (2020), especially the model configuration and design choices. There are a lot of similarities, including the use of FV3 (termed the FV3-SAR in that study, which I believe is the same model that is the core component of the UFS-SRW), variational and hybrid DA with GSI, similar physics choices, and a similar convective-storm case study approach (although for a different case). Can the authors describe how that work ties in with the current set of experiments?**

    - As mentioned by the reviewer, the main overlapping is regarding the use of the FV3LAM and the GSI analysis system. Other similarities that could be mentioned are

the model horizontal grid spacing of 3 km; numerical experiments executed assessing the hybrid weight of the ensemble background error covariance (BEC) with values of 0% (pure 3DVar), 75% and 100% (pure ensemble based BEC); similar microphysics parameterization (Thompson Aerosol-Aware; Thompson and Eidhammer (2014)); and similar convective case study. Nevertheless, our study greatly differs from Tong et al. (2020) in two main aspects: the direct assimilation of reflectivity data and the testing of the ensemble Kalman filter (EnKF) data assimilation method. In our study we assimilate radar radial velocity and the vertical azimuth display derived from radar radial velocity, but, we do not assimilate any reflectivity data. Meanwhile, we test functionalities in GSI such as the PBL pseudo-observations function and supersaturation removal that were not considered in Tong et al. (2020). Additionally, we focus on the hybrid 3DEnVar method in GSI since it is used in operational RAP and HRRR systems. For operational RRFS, development is underway to incorporate the EnKF into the hybrid data assimilation system for the first implementation. In order to account for the reviewer's comment, lines 110-113 were added to the manuscript and lines 90-93 were re-phrased.

Lines 90-93: "which studied the impact of the direct assimilation of radar radial velocity and reflectivity using the hybrid three dimensional ensemble–variational data assimilation (3DEnVar) and ensemble Kalman filter (EnKF) algorithms within the Gridpoint Statistical Interpolation (GSI; e.g., Wu et al., 2002; Kleist et al., 2009)."

Lines 110-113: "It is worth mentioning that despite some similarities with the work of Tong et al. (2020), in this study the focus is on the hybrid 3DEnVar method in GSI and configurations used in operational RAP and HRRR systems. For the operational RRFS, development is underway to incorporate the EnKF into the hybrid data assimilation system for its first implementation."

- **The RRFS will be an ensemble-based system, so generating and verifying ensembles seems like a good choice to assess the benefits of the various approaches. It may be useful to clarify why the authors only performed deterministic forecasts somewhere in the text (sorry if it's there and I missed it!).**

- The reviewer is right. Since RRFS is under development, RRFS v0.1 does not yet have the capability of generating ensemble forecasts. Lines 102-103 were added, as suggested.

Lines 102-103: "While single, deterministic forecasts are produced and evaluated in this study using RRFS v0.1, it should be noted that future RRFS implementations will produce convection-allowing ensemble forecasts."

- **I really think this study would benefit from additional cases, especially when the authors argue at many points in the paper they are using the results to guide future configuration decisions. Some of the differences between the experiments seem very small, and may become more evident with a larger sample size. This could be considered a "fatal flaw" by some, but I think there's**

**some merit in providing documentation of ongoing work leading up to the implementation of the future RRFS system in the form of this manuscript.**

- We agree with the reviewer. However, and as mentioned in the document, the intention of this study is to explore a variety of functionalities and configurations on a single case in order to establish an understanding of baseline sensitivities. Upon establishment of initial baselines, retrospective cases and real-time experiments spanning weeks and seasons will be examined as RRFS advances toward operational implementation of RRFS.

**Specific comments**
**Lines 40-42: This sentence implies that the addition of the WaveWatch model into the operational forecast somehow improved the low-level cold temperature bias observed in a prior version of the GFS. I don't think that's possible and I don't think the change notice referenced supports that claim. Please revise.**

- The reviewer is right. The sentence was revised as suggested.

  Lines 40-41: "Within the UFS framework, the GFS was coupled with the WAVEWATCH III wave model in the operational upgrade of March 2021 (NWS, 2021)"

**Line 87-91: I recommend removing these sentences. The number of studies that examine data assimilation for convection-allowing applications is too numerous to mention here, so describing these two specific studies is necessary, unless they are aspects of the work that are especially relevant to the current work.**

- The sentences were removed as recommended.

**Line 104: The authors should make clear that the eventual RRFS implementation will produce ensemble forecasts and not just a single deterministic forecast.**

- Lines 103-104 were added in order to clarify the point made by the reviewer.

  Lines 103-104: "Single deterministic forecasts are produced and evaluated in this study using RRFS v0.1, however, it should be noted that future RRFS implementation will produce convection-allowing ensemble forecasts."

**Line 153-161: I suggest moving the list of these parameterizations into a Table that can be referenced in the future, including names of schemes and associated studies that describe each scheme.**

- As suggested, the list of parameterizations was moved to Table 1. The text in lines 151-153 was adjusted accordingly.

  Lines 151-153: "The RRFS_PHYv1a suite is based on physical schemes implemented in the operational RAP, HRRR, and GFS systems and is used in all simulations in this study. Table 1 presents the RRFS_PHYv1a physics parameterizations and associated studies that describe each scheme, based on CCPP (2021)."

**Section 2.6: What do the authors mean by "workflow"? As written, the term is used rather generically, but I'm guessing that there is specific workflow software that is used that should be described in the text (this may be described later in the text, but the authors should bring this up earlier).**

- As recommended, the specific workflow software used was referenced earlier in the text in lines 205-207.

  Lines 205-207: ". It is based on the UFS SRW application v1.0.0 (UFS Development Team, 2021) community workflow which uses the Rocoto workflow management system (https://github.com/christopherwharrop/rocoto/wiki/Documentation)."

**Line 260-262: How were the MLCAPE and shear diagnostics computed? The text states they were "observed", but there are no routine soundings typically available between 19-20 UTC in northeastern Oklahoma. If these values are from a model analysis, that should be stated (e.g., "The RAP analysis contained MLCAPE values of…).**

- The reviewer is right. However, for this case study in particular, a sounding at 19 UTC is available in northeastern Oklahoma. The sounding with all the instability parameters can be accessed at https://www.spc.noaa.gov/exper/archive/event.php?date=20200504. Lines 255-256 were added to the document to clarify the source of the parameters.

  Lines 255-256: "The instability parameters are based on the observed sounding at 19:00 UTC over Norman, OKlahoma (KOUN)."

**Line 367: Are Oklahoma Mesonet observations assimilated?**

- Yes, as can be noted in Figure 4 by the + symbol.

**Line 543: Is model level 50 really located around 850 mb? Does that mean that there are 50 levels below 850 mb and only 14 levels above 850 mb (64 levels total)?**

- We thank the reviewer for pointing this out. In FV3LAM, the hybrid model levels are inverted from 0 to 64, thus, model level 64 is close to the surface and level 0 is at the model top. The hybrid model level 50 in this study corresponds then to level 15 which is located around 850 mb. Figure 15 and its caption as well as line 538 were modified accordingly. In addition, the vertical resolution of the model was modified since 65 vertical layers were used instead of 64 (line 267).

**Lines 554-556: How are the MMI values so different at 21 UTC in CLIPSAT and 75EnBEC? To my eye, the figures look almost identical. The differences at 23 UTC look more significant, but the MMI values are more similar at this time. Can the authors explain why this is the case?**

- We agree with the reviewer that in a visual and more subjective analysis the subplots for 21 UTC from both experiments look very similar (Figure 16a and d in the new version of the manuscript). In fact, that motivated the inclusion of the objective

analysis in the figures and results discussions showing the MMI (F+O) results for reflectivity values larger than 35 dBZ. In order to rule out any error in the results shown in this figure, the Method for Object-Based Diagnostic Evaluation (MODE) was re-applied but results were the same as shown in the manuscript. The figure below shows Figure 16 but only for forecasts of composite reflectivity above 35 dBZ with the corresponding contour of observed reflectivity at 35 dBZ. It can be noted that the experiment that produces more isolated objects which do not match with observed objects are penalized in the MMI values. This justifies the lower MMI (F+O) value presented in Figure 16d for the 75EnBEC experiment at 21 UTC. Results from the CLIPSAT experiment shows that more predicted reflectivity areas match better the observed one and it is reflected in the larger MMI (F+O) value obtained. At 23 UTC, it can be observed that the MMI (F+O) value of the 75EnBEC experiment has greatly increased when compared to the MMI (F+O) value at 21 UTC. Yet, the experiment CLIPSAT shows a better reflectivity forecast at this hour with a larger MMI (F+O) value. In CLIPSAT there is a reduction of the spurious convection between north-central Arkansas and south-central Missouri, but most of the spurious convection shown in 75EnBEC over other regions is also observed in CLIPSAT. This may be the reason why the differences are not very significant in the MMI values. To account for the reviewer's comment, lines 550-557 were added to the document and lines 626-628 were revised and modified.

Lines 550-557: "These values are greatly increased at 2 h forecast from 0.540 in 75EnBEC to 0.811 in CLIPSAT due to a better positioning and coverage of the reflectivity above 35 dBZ in areas over southeastern Missouri. Overall, more spurious convection over Missouri is shown in 75EnBEC which may have led to the lower MMI (F+O) at this forecast hour. At 4 h forecast, MMI (F+O) results show an increase from 0.698 in 75EnBEC to 0.793 in CLIPSAT, with a reduction of the spurious convection between north-central Arkansas and south-central Missouri in CLIPSAT. Nevertheless, most of the spurious convection shown in 75EnBEC over other regions is also observed in CLIPSAT, which may have penalized the MMI (F+O) values in the last experiment. At 6 h forecast, more similar MMI (F+O) values are found, but still less spurious convection is observed for lower reflectivity thresholds in CLIPSAT"

Lines 626-628: "At shorter forecast lead hours, it produces more skillful forecasts with a better positioning and coverage of the reflectivity above 35 dBZ and precipitation forecasts are as good as in experiments 75EnBEC and 100EnBEC."

[Figure]

2, 4, and 6 h forecasts of composite reflectivity from experiments CLIPSAT (a, b, and c) and 75EnBEC (d, e, and f) initialized at 19:00 UTC on 4 May 2020. Solid black lines are the 35 dBZ reflectivity observation contours, valid at the forecast time, respectively. MMI (F+O) results for reflectivity values larger than 35 dBZ are shown in the lower right corner of each panel.

**Figure A1: Why is this included as figure A1 and not Figure 11?**

- As indicated, Figure A1 was removed from the Appendix and included as Figure 11 in the new version of the manuscript (see lines 489, 491, and 492). The figures below were renumbered accordingly.

---

## Author Response (AR3)

Ivette Hernández Baños
National Center for Atmospheric Research
3450 Mitchell Ln, Boulder, CO 80301
ivette@ucar.edu

4 May 2022

Dear Topical Editor Travis A. O'Brien,

Thank you for your comment on the manuscript. Our response to your one remaining minor concern is noted below. Changes to the document are highlighted in blue.

Best regards,
Ivette on behalf of the authors

**The one remaining minor concern is associated with the most recent reviewer's final comment: "How are the MMI values so different at 21 UTC in CLIPSAT and 75EnBEC?" The additional text and figure (provided in your response) does help explain what a reader should be looking for in the figures, but I still am having difficulty seeing how the 75EnBEC experiment "produces more isolated objects which do not match with observed objects," such that it has a penalized MMI value. From my visual assessment, it appears that there are just as many non-matching, isolated cells with reflectivity greater than 35 dBZ in CLIPSAT as there are in 75EnBEC. Can you and your co-authors come up with a way to visualize the differences in the two simulations such that it is obvious that CLIPSAT should have a substantially higher MMI? Perhaps there is some way to highlight the convective regions in both simulations that detract from the MMI score?**

We thank the editor for the opportunity to revisit this issue. We recognize that indeed it is not very easy to see in Figure 16d how many more isolated objects were predicted in the 75EnBEC experiment that did not match with the observed objects. In an attempt to better show the differences between the experiments CLIPSAT and 75EnBEC at 21 UTC (2 h forecast), Figure 1 presents the index of the objects identified using MODE for the forecast and observed reflectivity values larger than 35 dBZ in both experiments. A total of 4 objects were identified in the observed field while 7 forecast objects were identified in CLIPSAT and 10 forecast objects in 75EnBEC. Object number 1 is not shown in the figure since it falls far from the interest area but its statistics are shown in Table 1. Table 1 shows the maximum of the total interest for the observed (MIO) and forecast (MIF) objects for both experiments. Overall, in Figure 1 it can be noted that in both experiments there were matched (colored red indices) and unmatched (colored royal blue indices) objects, with forecast objects 9 and 10 in 75EnBEC (Figure 1b) and 5 and 7 in CLIPSAT (Figure 1c) being unmatched over northeastern Oklahoma, southeastern Kansas, and southwestern Missouri. This means that both experiments overforecast the reflectivity values of the convection over that area. As a result, total interest values from these unmatched objects penalized the MMI (F+O) values of both experiments as can be noted in Table 1, where 0.4233 and 0.29 are the MFI of forecast objects 9 and 10 in 75EnBEC, respectively, and 0.5355 and 0.3057 are the MFI of forecast objects 5 and 7 in CLIPSAT, respectively. Yet, in 75EnBEC forecast object number 2 was also identified, which did not match any of the observed objects and resulted in the lowest MFI in this experiment with a value of 0.2339. Nevertheless, the best MIF and MIO value of 0.9532 is found in 75EnBEC for the observed object number 2 matching forecast object 5. A similar MIF and MIO value of 0.9472 was obtained for forecast object number 3 in CLIPSAT. It

is worth noting that observed object 3 is best matched by forecast object number 6 in CLIPSAT with an MIF of 0.9075 while its best match in 75EnBEC is with forecast object number 8 with an MIF of 0.7429. The identified forecast object number 3 in 75EnBEC also penalized the MMI (F+O) with an MIF of 0.3103. Finally, the median of all MIF and MIO of each experiment indicates that, overall, the CLIPSAT experiment performs better than 75EnBEC at this forecast hour.

[Figure]

Figure 1. Index of the objects identified by MODE for the observed reflectivity (a) and 2 h forecast of composite reflectivity in experiments 75EnBEC (b) and CLIPSAT (c) for values larger than 35 dBZ. The colored red indices indicate matched objects between the forecast and observation and colored royal indices represent the unmatched objects.

Table 1. Maximum of the total interest derived from the comparison between an object in one field (observed or forecast) with all objects in the other field (observed or forecast).

| | | 75EnBEC | | | CLIPSAT | | |
|---|---|---|---|---|---|---|---|
| | Obj # | MIO | MIF | MIO+MIF | MIO | MIF | MIO+MIF |
| | 1 | 0.85 | 0.3378 | 0.85 | 0.811 | 0.3378 | 0.811 |
| | 2 | 0.9532 | 0.2339 | 0.9532 | 0.9472 | 0.4124 | 0.9472 |
| | 3 | 0.7429 | 0.3103 | 0.7429 | 0.9075 | 0.9472 | 0.9075 |
| | 4 | 0.504 | 0.5128 | 0.504 | 0.4682 | 0.811 | 0.4682 |
| | 5 | | 0.9532 | 0.3378 | | 0.5355 | 0.85925 |
| | 6 | | 0.5678 | 0.2339 | | 0.9075 | 0.3378 |
| | 7 | | 0.8204 | 0.3103 | | 0.3057 | 0.4124 |
| | 8 | | 0.7429 | 0.5128 | | | 0.9472 |
| | 9 | | 0.4233 | 0.9532 | | | 0.811 |
| | 10 | | 0.29 | 0.5678 | | | 0.5355 |
| | | | | 0.8204 | | | 0.9075 |
| | | | | 0.7429 | | | 0.3057 |
| | | | | 0.4233 | | | |
| | | | | 0.29 | | | |
| MMIO | | 0.79645 | | | 0.85925 | | |
| MMIF | | | 0.46805 | | | 0.5355 | |

| | | | | | |
|---|---|---|---|---|---|
| **MMI (F+O)** | | | **0.5403** | | **0.811** |

As the editor suggested, Figure 16 in the manuscript was updated including a blue circle in panel (a) and a red circle in panel (d) highlighting the area of forecast objects 2 and 3 that detract from the MMI score. Lines 549-550 and 553-555 were also modified. More information on how MODE works in MET can be found at https://met.readthedocs.io/en/main_v10.1/Users_Guide/mode.html#mode-output and it is recommended to refer to Davis et al. (2009) (https://doi.org/10.1175/2009WAF2222241.1) for more information on the MMI metric.

Lines 549-550: "Both experiments overforecast the reflectivity values larger than 35 dBZ over that area, though."

Lines 553-555: "Overall, more spurious convection over northwestern Missouri is shown in 75EnBEC which led to the lower MMI (F+O) at this forecast hour (see the blue and red circles over this area in Fig. 16a and d, highlighting improvement and degradation, respectively, for each experiment)"

Figure 16:

[Figure]

**Figure 16.** As in Fig. 7, but for experiments CLIPSAT (a, b, and c) and 75EnBEC (d, e, and f). The red (blue) circle in panel d (a), respectively, indicates forecast convection that penalized (improved) the MMI (F+O) value in experiment 75EnBEC (CLIPSAT) at 2 h forecast.